# Identifying the seeding signature in cloud particles from hydrometeor residuals

Mahen Konwar[1*], Benjamin Werden[2,], Edward C. Fortner[2], Sudarsan Bera[1], Mercy Varghese[1], Subharthi Chowdhuri[1,&], Kurt Hibert[3], Philip Croteau[2], John Jayne[2], Manjula Canagaratna[2], Neelam Malap[1], Sandeep Jayakumar[1], Shivsai A. Dixit[1], Palani Murugavel[1], Duncan Axisa[4], Darrel Baumgardner[5], Peter F. DeCarlo[6], Doug R.Worsnop[2], and Thara Prabhakaran[1]

[1] Indian Institute of Tropical Meteorology, Ministry of Earth Sciences, Pune, India 411008

[2] Aerodyne Research Inc., Billerica, MA, USA, 01821

[3] Weather Modification Inc., Fargo, ND, USA, 58102

[4] Center for Western Weather and Water Extremes, Scripps Institution of Oceanography, La Jolla, CA 92037, USA

[5] Droplet Measurement Technologies, Longmont, CO, USA, 80503

[6] Department of Environmental Health and Engineering, Johns Hopkins University, Baltimore, MD USA 21218

[&] now at University of California, Irvine, CA 92697-2700, USA

*Corresponding author

Dr. Mahen Konwar

Indian Institute of Tropical Meteorology

Dr. Homi Bhabha Road, Pune 411 008, India.

Email: mkonwar@tropmet.res.in

**Abstract:**

Cloud seeding experiments for modifying clouds and precipitation have been underway for nearly a century; yet practically all the attempts to link precipitation enhancement or suppression to the presence of seeding materials within clouds remain elusive. In 2019, the Cloud-Aerosol Interaction and Precipitation Enhancement Experiment (CAIPEEX) investigated residuals of cloud hydrometeors in seeded and non-seeded clouds with an airborne mini-Aerosol Mass Spectrometer (mAMS). The mAMS was utilized in conjunction with a counterflow virtual impactor (CVI) inlet with a cutoff diameter size of approximately 7 μm. The evaporated cloud droplets from the CVI inlet as cloud residuals were evaluated through the mAMS. The Chlorine (Cl) associated with hygroscopic materials, i.e., Calcium Chloride ($CaCl_2$) and potassium (K), which serve as the oxidizing agents in the flares, is found in relatively higher concentrations in the seeded clouds compared to the non-seeded clouds. In convective clouds, Cl and K as cloud residuals were found even at an distance 2.25 km from the cloud base. Major findings from the seeding impact are: an increase in the number concentration of small (<20 μm) droplets and an indication of raindrop formation at 2.25 km above the cloud base. It is demonstrated that the seed particle signature can be traced inside clouds along with the microphysical impacts.

## 1. Introduction:

E.G. Bowen first proposed in 1952 that hygroscopic particles can foster collision-coalescence (CC) processes in a cloud (Bowen, 1952). Since then, cloud seeding experiments have been conducted worldwide to mitigate and respond to the ever-increasing urban water demand during a drought season or in drought-prone regions. More than 50 countries are involved in weather modification projects (Flossmann, et al., 2019). Over the years, the interest in rain enhancement projects has increased due to the accumulating evidence of a potentially positive effect (i.e., enhancement in rainfall) in several seeding experiments (Mather et al., 1996; Mather et al. 1997; Bruintjes, 1999; WMO, 2000; Gayatri et al., 2023; Prabhakaran et al., 2023). However, skepticism remains within the broader cloud physics community because the efficacy of many cloud seeding experiments remains inconclusive (Ryan and King, 1997; Silverman, 2003; Flossmann et al., 2019). In addition to the existing challenges of evaluating the effectiveness of cloud seeding experiments, other pivotal longstanding issues revolve around accurately detecting the hygroscopic particles released within a cloud, identifying the seeded cloud, and comprehending the impact of seeding on the cloud microphysical properties.

Traditionally, in a cloud seeding experiment tracers such as the inert gas, sulfur hexafluoride ($SF_6$) (Stith, et al., 1986; Stith et al., 1990; Bruintjes et al., 1995; Rosenfeld et al., 2010), or radar chaff at cloud bases are released, and then efforts are made to measure these tracers higher in the cloud. However, tracing of $SF_6$ in a seeded cloud is challenging and successful trials have been reported only on a few occasions near the cloud base (Rosenfeld et al., 2010). The detection of $SF_6$ and chaff traces is hampered by detection limits, especially in the presence of high background concentrations. Using these tracers as proxies for tracking air masses carrying seeding material is limited by the challenge of unambiguously connecting their presence with the seeding material due

to their non-reactive nature with cloud particles. Consequently, several questions arise during these
experiments. For instance, does the dispersed seeding material effectively enter the targeted cloud
region? Up to what altitude do these materials reach? Are the in-situ measurements being conducted
within the intended cloud volume? How can transported flare particles be located within large
clouds? Due to these uncertainties the need to more quantitatively evaluate the direct link between
seeding materials and the formation of cloud hydrometeors, the development of a low-impact but
more effective tracer has been recommended, e.g. Tessendorf et al., (2012).

81       A critical question in any cloud seeding experiment is whether the observed changes in the

cloud microphysical properties after seeding are due to the introduction of seeding material or to
natural cloud processes. There are two requirements necessary to address this question: (i) Can the
trajectory of seeding material be successfully traced in the cloud, and (ii) can changes in cloud
microphysical processing be linked to seeding materials? In this study, an instrumented aircraft was
deployed to acquire convincing evidence that addresses these questions. This work primarily
addresses how to trace seed particles' signatures in clouds and focuses on the question of changes in
cloud micrpphysical properties due to the introduction of seeding particles. This novel technique
uses a mini-Aerosol Mass Spectrometer (mAMS) (Jayne et al., 2000) behind a counterflow virtual
impactor (CVI) (Noone et al., 1988; Shingler et al., 2012) to identify seeding material in the cloud
droplets residuals i.e., the aerosols that remain after evaporation of the cloud droplets.

92       The hygroscopic cloud seeding hypothesis relies on a chain of microphysical  processes.

Dispersal of giant cloud condensation nuclei (CCN), hygroscopic particles with diameter  between 1-
10 μm, in the updraft region of cloud base adds larger drops to the tail of the natural cloud droplet
size distribution (DSD), known as the 'tail effect'. This effect further accelerates the formation of
raindrops through CC (Segal et al., 2004; Segal, et al., 2007; Kuba and Murakami, 2010; Konwar et

al, 2023). With the initial activation and growth of these larger CCN, the supersaturation over water droplets ($SS_w$) decreases above the cloud base. As a result, the smaller, natural CCN do not activate. This effect reduces the total droplet number concentration ($N_t$, $cm^{-3}$) and broadens the DSDs, a phenomenon known as the 'competition effect' . This broadening fosters the droplet growth rate by intensifying the CC process, which accelerates the formation of precipitation (Cooper et al., 1997; Rosenfeld et al., 2010). Past studies used in-situ measurements to evaluate well-formed seeded clouds whose formation revealed a broadening of the DSDs by hygroscopic seeding in marine stratocumulus clouds (Ghate et al., 2007). Researchers reported that an increased concentration of small cloud droplets occurred at an earlier stage, while at a later stage, an increased concentration in the large  size range of 20-40 μm was noted. In another study, $SF_6$ was used to track  air parcels in a seeded cloud, where milled salt particles were used as the seeding agent. In this study a broadening of the DSD was observed (Rosenfeld et al., 2010). Linking the evolution of cloud microphysical processes to hygroscopic seeding remains elusive despite worldwide hygroscopic cloud seeding experiments (Flossmann et al., 2019; Silverman 2003; Tessendorf et al., 2012). The major hurdle is that the physical processes leading to precipitation formation are dynamic and complex and difficult to  directly and quantitatively track and link to the seeding (Tessendorf et al., 2012).

In the current study, using an mAMS, we demonstrate that the seeding signatures within stratus and convective clouds are detectable with an evidence-based approach without using tracer gasses. We further show that the seeding materials and the seeding-activated cloud droplets in convective clouds can propagate to higher altitudes while also modulating the cloud's microphysical properties. The ultimate goal is to investigate the microphysical pathways that are modified in cloud seeding operations. These experiments took place in the region near Solapur (17.66° N, 75.90° E),

India, during the Cloud-Aerosol Interaction and Precipitation Enhancement Experiment (CAIPEEX)
(Prabha et al., 2011; Kulkarni et al., 2012; Prabhakaran et al., 2023) in 2019 (phase-IV).
**2. Materials and Methods:**
**2.1 Measurements of cloud properties.**
Three cloud seeding events carried out on 21 August, 23 August and 24 August in 2019,  are
selected here for evaluation of seeding signatures and plausible links to microphysical properties.
Instruments for the measurement of flare particles, aerosol, and cloud properties were operated on a
Beechcraft-B200 aircraft. This aircraft was equipped with flare racks located under both the wings
and the belly. The flare racks in the wings are used for warm cloud seeding operations (Mather et al.,
1997), while the belly is utilized for cold cloud seeding  operations (French et al., 2018; Friedrich et
al., 2020). The temperature (T, ºC), relative humidity (RH%), wind speed (ms$^{-1}$) and wind directions
were measured with the Airborne Integrated Meteorological Measurement System (AIMMS-20).
The DSD in the size range of 2-50 μm was measured with a Cloud Droplet Probe (CDP-2)
manufactured by Droplet Measurement Technologies LLC, USA. The bulk microphysical properties
are derived from the measured DSDs, e.g. the total number concentration ($N_t$, cm$^{-3}$) and liquid water
content (LWC, g m$^{-3}$). The effective radius ($r_e$, μm) was calculated from the ratio between the third
and second moments of the DSDs (Martin et al., 1994). The Precipitation Imaging Probe (PIP) was
used to document drizzle drops in the cloudsover the size range of 100-6200 μm. The technical
specifications of these instruments are shown in Table 1. The uncertainties associated with the CDP,
and single particle light scattering instruments like the CDP, have been well characterized and
documented (Baumgardner et al., 1983, 2001, 2016; Lance et al., 2010). In water droplets the sizing
uncertainty is ±20% and counting accuracy ±16%, which propagates into a LWC uncertainty of

141  ±38%.

Cloud properties are altered by the entrainment of cloud-free air masses at the edges of the
cloud; hence to minimize the influences of entrainment and mixing processes in the seeded and non-
seeded clouds, only clouds with near adiabatic or slightly diluted cloud parcels are considered to
evaluate cloud microphysical properties. Only cloud passes with LWC in the range of $0.75 <$
$LWC/LWC_{max} < 1$ (Konwar et al., 2021) were selected for this study. Here, $LWC_{max}$ represents the
maximum measured value of LWC during a cloud pass. Note that this cloud regime may be
considered as the cloud core, typically located within the strongest updrafts zone. Our main aim is to
select the DSDs located within the cloud core regime. Note that in most naturally developing clouds
the $LWC_{max}$ values are less than the adiabatic LWC ($LWC_{ad}$) values because of the entrainment of
drier air, mixing, precipitation fallout and radiative heating/cooling (Korolev et al., 2007). The
maximum adiabatic fraction, $AF_{mx}=LWC_{max}/LWC_{ad}$, indicates the extent of dilution that has
occurred in the cloud core regime. During their development and dissipation stages clouds undergo
significant changes; therefore, it is practically impossible to find two clouds identical in all states, let
alone their lifetimes. It is to be noted that the AF values may not accurately represent the mixing
state when CC is significant and drizzle particles form within the clouds. Additionally, studies of the
seeding effect using parcel model simulations without the inclusion of mixing processes indicates a
significant change in the LWC profile compared to the non-seeded cloud (Konwar et al., 2023). Such
changes in LWC values at different vertical distances from the cloud base of the seeded clouds do
not necessarily imply the true dilution rate in the observations. Since the cloud seeding flare
produces high concentrations of small-sized particles, they can be activated into cloud droplets in
strong updraft regimes with high supersaturation (Konwar et al., 2023; Prabhakaran et al., 2023). In

163 a parcel model simulation, small aerosols released from flares are found to be activated due to an

164 increase in supersaturation when the collision-coalescence process is active (Konwar et al., 2023).

165 For details on the nucleation process within the zone of intense collision, where rapid decrease in

166 drop concentration leads to an increase in supersaturation, readers are referred to Pinsky and Khain

167 (2002). At a given height, however, seeding does not change the adiabatic value, but activation of

168 new particles at a given level due to seeding can alter the AF. Another aspect is that near the cloud

169 base the $LWC_{ad}$ values are quite small (e.g., $< 1$ g m$^{-3}$), therefore any small change in the measured

170 LWC could indicate a large change in AF. With this background information in mind, the DSDs for

171 Seed Cloud (SCl) and No Seed Cloud (NSCl) conditions are compared at different vertical distances

172 above the cloud base (D*, km). The lowest unbroken visible section of a convective cloud was

173 selected as the cloud base. The cloud top is defined as the maximum altitude attained by these clouds

174 at any given moment during their development.



**Table 1**

Details of Instruments used on the aircraft and for offline analysis in the study

| Instrument | Variable | Range/Remark | Reference |
|---|---|---|---|
| Aventech AIMMS-20 | GPS Coordinates, altitude above Mean Sea Level (MSL), temperature, dew point temperature, horizontal and vertical winds | Vertical wind accuracy 0.75 m s$^{-1}$ | https://aventech.com/products/aimms20.html |
| DMT CDP2 | Cloud droplet number concentration and size distribution | 3.0 – 50.0 µm | https://www.dropletmeasurement.com/product/cloud-droplet-probe/ |
| DMT PIP | Particle image | 100 µm – 6.2 mm | https://www.dropletmeasurement.com/product/precipitation-imaging-probe/ |
| CVI | Droplet/ice crystal residuals | Particle Cut size ~ 7µm | https://www.brechtel.com/product/aircraft-based-counterflow-virtual-impactor-inlet-system-cvi/ |





**2.2 Measurement of hygroscopic flare particles by mAMS and Correcting time trends of slow-vaporizing species**

We utilized a mAMS to analyze the chemical compositions of residual particles from cloud droplets, specifically to trace flare particles within the seed clouds. The CVI is manufactured by Brechtel Manufacturing Inc. (BMI, Model 1204, www.brechtel.com). The cloud droplets were passed through the CVI to obtain the droplet residual that were sampled by the mAMS. Through the use of inertial impaction, the CVI inlet allows cloud hydrometeors with aerodynamic diameters larger than a certain size to pass through, depending on the velocity of the counterflow. A warm, particle-free dry nitrogen gas is directed towards the inlet against the direction of the ambient air flow. This causes a separation of in the incoming free stream air, with particles >7 μm in the sampled air having enough inertia to penetrate the counterflow and join the sample flow. The CVI adjusted flow rates with its internal software based on true air speed (TAS) obtained from the AIMMS. The cut-size is a function of various factors, e.g., air pressure, air speed, and the average angle of attack, is known to have an uncertainty of approximately ±1 μm. The heated air evaporates cloud droplets and the remaining dried residuals enter the mAMS where their chemical compositions are classified. Details of the operational principles of the CVI can be found in Ogren et al., 1985; Ogren, 1987; Noone et al., 1988; Shingler et al., 2012; Golderger et al. 2020; and references therein.

The mAMS measured the residual particles with vacuum aerodynamic diameters of less than 1 μm, sampling through an aerodynamic lens. The aerosol sample stream is intermittently blocked to measure background signals. The aerosol signal is the difference between unblocked ("open") measurements and those obtained during the blocked ("closed") period. The mAMS sampled 10 seconds of closed signal for every 110 seconds of open. The heater, operated at 600 $^{o}$C, vaporized the sample, electron impact ionized the vapors, and the resultant ions were extracted into the mass

analyzer for measurement of chemical composition and mass distributions (Jayne et al., 2000;
DeCarlo et al., 2006; Canagaratna, et al, 2007; Drewnick et al., 2015; Giordano et al., 2018; Salcedo
et al., 2006).

209        Ice Crystal Engineering (ICE) Inc. (USA) manufactured the hygroscopic flares used in this

work. The flares were composed of an aggregated mixture of potassium perchlorate ($KClO_4$) and
calcium chloride ($CaCl_2$) (Hindman, 1978; Bruintjes et al., 2012).

212        For non-refractory ambient aerosol species (i.e., $NH_4$, $NO_3$, $SO_4$) aerosol concentrations are

obtained from the difference between the open and closed signals. The vaporization of non-
refractory aerosol species at 600°C typically completes on the timescale of hundreds of
microseconds, however, semi-refractory species such as metals and salts may take minutes to
completely vaporize (Canagaratna et al., 2007; Salcedo et al., 2006).

217        As discussed below, the Cl, HCl, and K from the $KClO_4$ and $CaCl_2$ in flares is a semi-

refractory species which exhibits slow vaporization. These slow vaporizing species were analyzed
using only the open signals.The background signal was calculated from measurements obtained
immediately before the cloud intercept of interest.

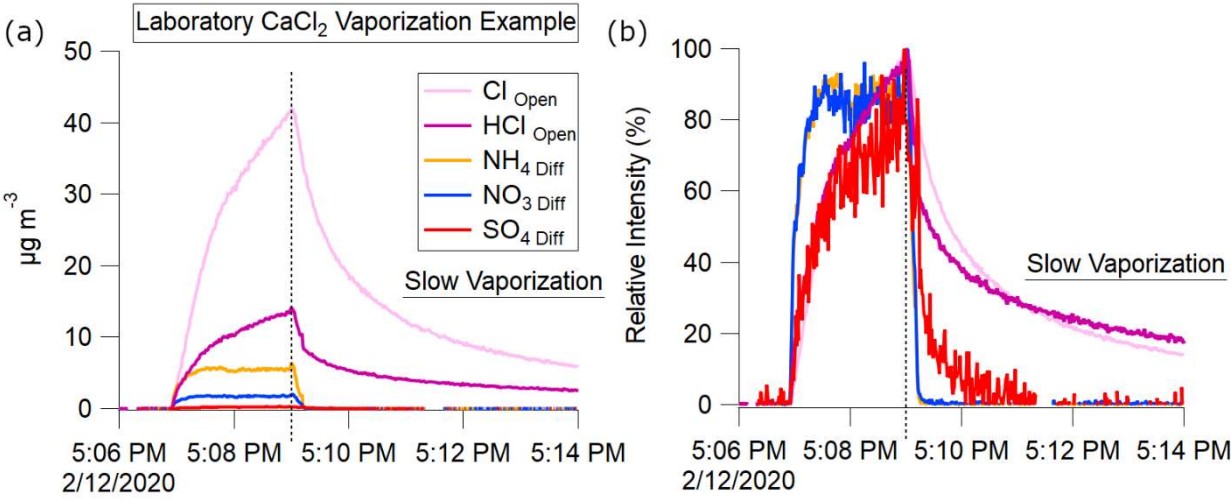


**Figure 1**. Laboratory atomized $CaCl_2$ AMS measurements observing slow vaporization of semi-refractory Cl species on 2/12/2020. Atomization begins at 5:07 PM ending at 5:09 PM. Slow vaporization is evident after 5:10 PM. The presence of $NO_3$, $NH_4$, and $SO_4$ are from calibration species ($NH_4NO_3$, $NH_4SO_4$) contaminants in the atomizer.

CaCl$_2$, the seeding component in the flares, has a melting point of 774 ºC. Laboratory measurements of atomized $CaCl_2$, primarily detected as Cl and HCl ions, exhibit the same slow vaporization seen in refractory salts (Drewnick et al., 2015). Fig. 1 shows a comparison of vaporization timescales of $CaCl_2$, $NH_4NO_3$, and $(NH_4)_2SO_4$ obtained with an AMS during laboratory measurements of $CaCl_2$ in solution with $H_2O$ which had been atomized and passed through a drier before sampling. This behavior differs from that observed from non-refractory $NH_4NO_3$ and $(NH_4)_2SO_4$, which were present as tracers.

233

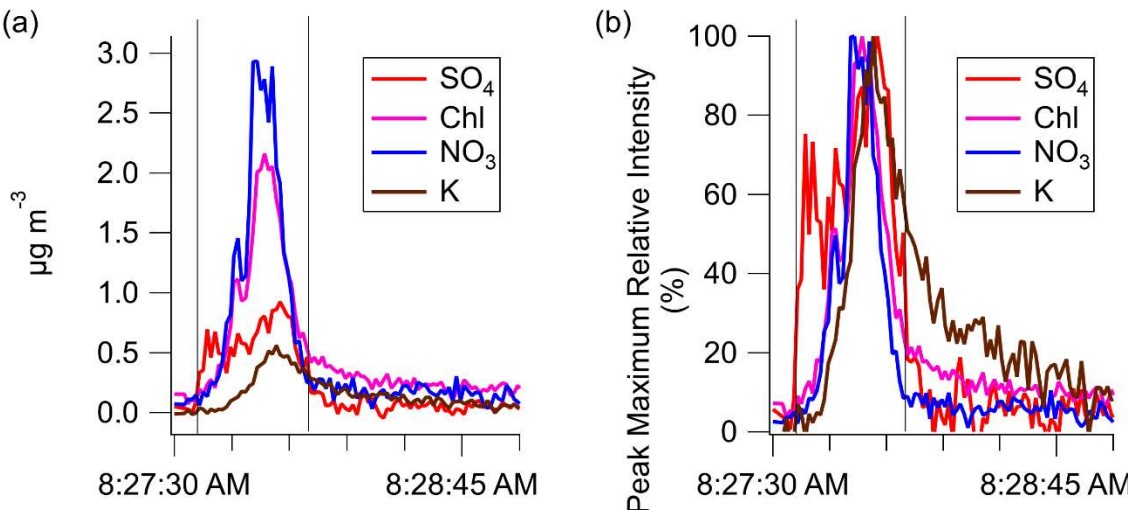

234

**Figure 2**. (a) shows the slowed time response of thespecies K and Cl for a seeded cloud pass on

August $23^{rd}$  (b) the relative intensity with respect to peak maximum of each species highlights the

slowed decay of K and Chl compared to $SO_4$ or $NO_3$.

The seeded cloud pass shown in Fig. 2a illustrates a single seeded cloud pass. The K and Cl

time series have a delayed decay to background compared to sulfate or nitrate. The relative intensity

shown in Fig. 2b highlights the delayed response in the decay of the two flare associated species (K,

Cl).

An exponential decay was fit to each cloud intercept, from the signal peak to 5 e-folding

times. The average decay exponential($\tau$) for Cl, and K across all seeded cloud intercepts, is shown in

Table 2.




**Table 2**

Average decay time constants from seeded cloud intercepts during CAIPEEX- IV, 23 August 2019.

| T | K | HCl | Cl |
|---|---|-----|----|
| Mean | 6.7 | 3.4 | 3.3 |
| Std Dev | 2.3 | 0.5 | 0.8 |


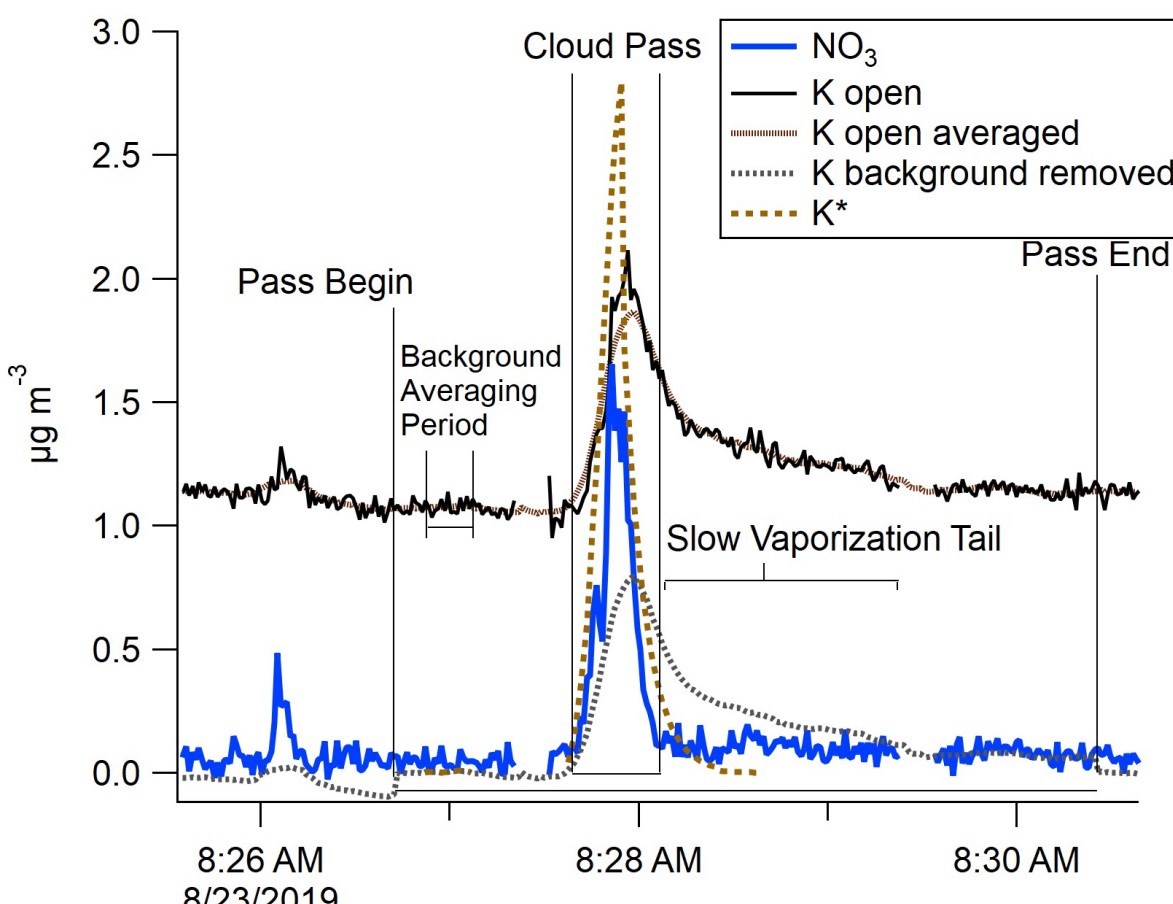


**Figure 3**. The measured semi-refractory open K signal and corrected K* signal from the mAMS are
depicted for a seeded cloud pass on 23 August 2019. The periods from the beginning to the end of
the cloud passes are also shown.

For each slowly vaporizing species, a new corrected time series was created. The start, stop, and
maximum total mass times were identified for each cloud pass (Fig. 3). For each species, a
background signal was determined from measurements during the non-cloud period preceding each
pass. This background was subtracted from the signal observed during each cloud intercept.
The cloud intercept time series peakat the same time as the uncorrected series. However, the tails
were corrected to decay within 5 tau e-folding times, while preserving the total mass. The equations
used in these calculations are shown below.
The measured mass from the start of the pass to the end of the slow vaporization regime was scaled
by the ratio of the total area divided by the area of fast vaporization (equation 1)
$Conc._{Areacorrected}(t)\big|_{End+(5\tau)}^{Start} = (Conc.(t) - Conc._{Background}) * \frac{Area_{Peak+Tail}}{Area_{Peak}}$    (1)
The decay of this normalized mass is adjusted to the exponential decay fit (Table 2) to the slow
vaporized mass (equation 2). This decay extends from the cloud pass peak to the end of the normal
vaporization period plus five e-folding times (Giordano et al., 2018)
$Conc._{TailCorrected}(t)\big|_{End+(5\tau)}^{Peak} = Conc._{AreaCorrected}(t) * e^{(-(\frac{1}{\tau})t)}$    (2)
This decay-corrected time-shifted time series is normalized to the unmodified slow vaporizing total
mass (equation 3)
$Conc._{Corrected}(t)\big|_{End}^{Start} = Conc._{TailCorrected}(t) * \frac{Area_{Peak}}{Area_{Peak} + Are_{Peak+Tail}}$    (3)

Finally, we applied an enhancement factor correction to the mAMS data resulting from the ambient
aerosol concentration being concentrated in the CVI by following Shingler et al., (2012).

## 3. RESULTS

### 3.1.1 Slow vaporization of semi-refractory seed aerosols

Although many aerosol species readily vaporize at 600 $^{\circ}$C, some semi-refractory materials in nature do not. Submicron aerosol particles in the troposphere, that contain Cl, are rarely semi-refractory and vaporize quickly in the mAMS. However, Cl in seeded clouds was found to vaporize slowly. The Cl measured in clouds seeded using $CaCl_2$ and $KClO_4$ exhibited the same slow vaporization (Fig. 2) as Atomized $CaCl_2$ in the laboratory (Fig 1). The majority of atmospheric Cl-containing aerosols are non-refractory. In our study the slowly vaporizing Cl was only observed in seeded clouds; thus, we assume that the source of the slow vaporizing Cl was from the flare material. Aerosol K is uncommon except as super micron mineral dust. As shown in Fig.2b, slowly vaporizing signals of Cl and K were observed in the campaign during seeded cloud intercepts.

The combination of the isolation of cloud residuals by the CVI and the presence of K and semi-refractory Cl allow for discrimination of the particles containing the flare combustion products.

The element Ca, was also present in the flare. The boiling point of Ca of 1484 $^{\circ}$C at ambient pressure means that this species was not vaporized inside the AMS and is thus considered a refractory species. Since Ca could not be observed in our study, the focus remained on the other species present.

As previously discussed, the time series of semi-refractory Cl and K signals are corrected to account for the difference in the decay response of slowly vaporizing species in the mAMS. Fig. 3 depicts the corrected (K*) and uncorrected semi-refractory K signals in the mAMS measurements for a seeded cloud pass, defining the periods for the start, peak, end, and tail of the pass.




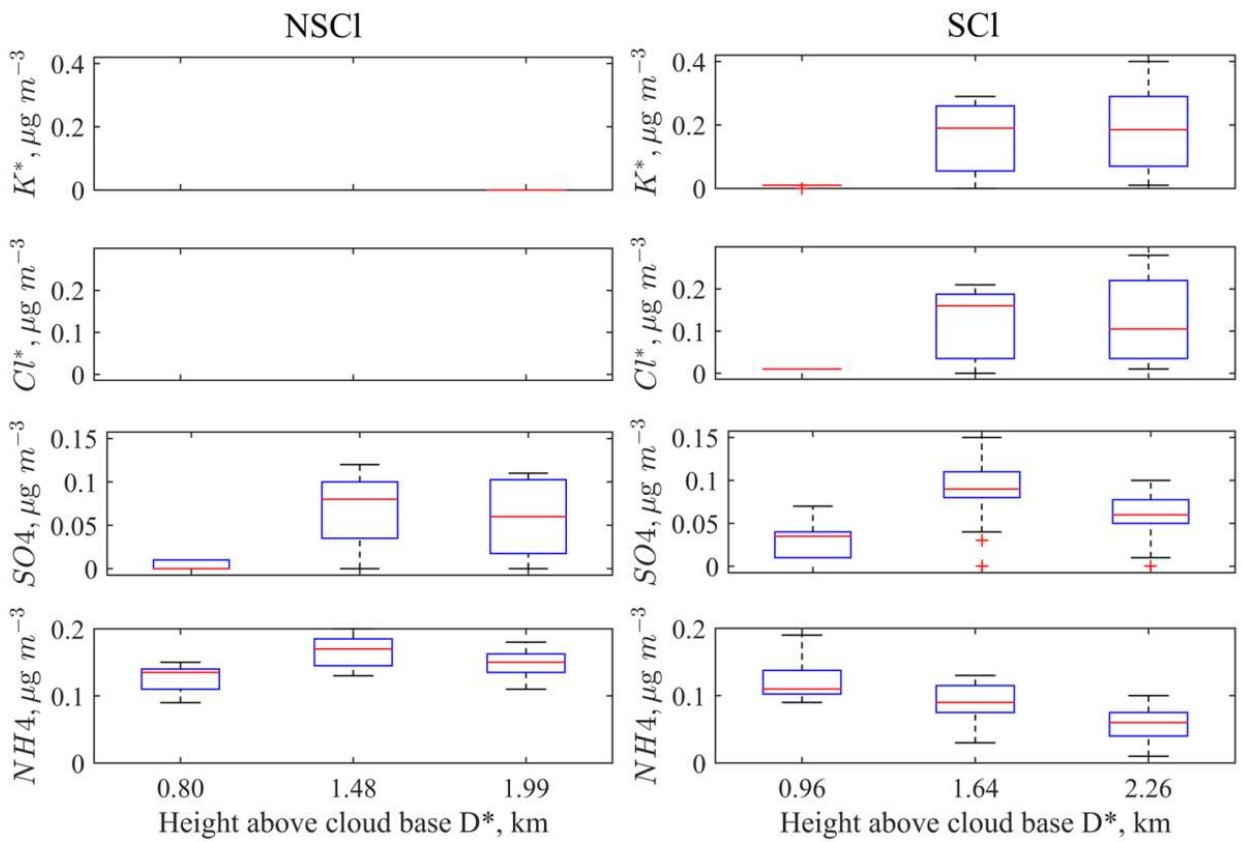


**Figure 4**. mAMS measurements of the mass concentrations of $Cl^*$, $K^*$, $NO_3$, and $SO_4$ versus D*
(km) for cloud particle residuals from six cloud passes through the same cloud on 23 August 2019.
The vertical profile box plots of each mAMS species at different altitudes shows median
concentration and range (25-75[th] percentiles).Three non-seeded clouds (NSCl) and three seeded
clouds (SCl) are shown.

A vertical profile of cloud residual aerosols, within the same cloud, taken before and after
seeding, provides a platform for measuring and observing cloud physical and chemical changes. The
resultant mAMS measurements from one such experiment, on August 23, 2019, with three cloud
passes of the same cloud before and three passes after seeding are shown in Fig 4.
In the mid level, all chemical species were found in higher quantities in the seeded cloud than in the
non-seeded cloud. Cl and K concentrations were significantly increased for all seeded cloud passes
compared to non-seeded cloud passes. The measurement of the flare chemical species in the seeded
cloud indicates that the mAMS could successfully identify the cloud droplets that containing
seeding material.
An additional observation is the increased $NO_3$ and $SO_4$ concentration in the cloud drops of seeded
clouds at upper heights. We hypothesized that the increased concentrations of these two chemical
species could be linked with the activation of the flare particles and other organics while mixing with
the naturally available $NO_3$ and $SO_4$ aerosols. The increased concentration of $NO_3$ in the seeded
cloud may also be due to the presence of more LWC. The additional water drives nitric acid ($HNO_3$)
from gas to liquid $NO_3$ (Wang and Laskin, 2014).
This example highlights the ability of the mAMS to identify flare associated species, by both
increased concentration and time response, in order to confirm the presence of  seeding material in
cloud droplet residual.

## 3.2 Seeding experiment, Seeding Signature, and Cloud properties

**3.2.1. Case i: 21 August 2019**. The flight pattern of the aircraft during  the cloud seeding
experiment conducted on 21 August 2019 in a warm stratus layer is shown in Fig. 5a. The objective
was to identify the seeding materials and record the cloud microphysical properties. The wind
direction was north-westerly at an altitude of nearly 4.10 km with a mean wind speed of 7 ms$^{-1}$.
Cloud passes (T=5.14 ºC, H=4.39 km) were made through the stratus layer before the dispersal of
seeding materials. Four hygroscopic flares were burned, two at a time, inside the layer cloud, from
8:01-8:08 UTC at H=4.10 km. Weak updrafts (W=0.61±1.53 m s$^{-1}$) prevailed indicate that the flare
material might have drifted horizontally. Increased mass concentrations of K* and Cl* are noted in
the downwind after the dispersal of the seeding agents, as shown in Fig. 5b and 5c. Repeated
crosswind cloud passes at a similar level (T= 6.44 ºC, H= 4.10 km) were made downwind of the
seeding. The aircraft could release non-volatile and fine aerosol particles through exhaust emission
(Anderson et al., 1998), which may also contaminate the cloud mass. Prabhakaran et al. (2023)
measured aerosol size distribution of background airmass, and then the background with aircraft
exhaust during CAIPEEX. They reported that the aircraft exhaust can impact mean radius, spectral
width and number concentrations of different modes of log-normal aerosol size distribution (see the
supplementary materials at https://doi.org/10.1175/BAMS-D-21-0291.2). Solution of simple
advection equations indicates dispersal of seeding plumes in the  downwind region after nearly 3
minutes (not shown here) where the aircraft also recorded enhanced concentrations of K$^{*}$ and Cl$^{*}$.
Gayatri et al., (2023) illustrated the seeding impact downwind of the seeded area through the high-
resolution numerical model in similar monsoon environment with the monsoon low-level jet (LLJ)
as detailed in the present study. The cloud bases are situated very close to the region with high wind
speeds in the monsoon low-level jet and the advection of seeding plume downwind of the seeded
location is noted. However, the fact that seeding was done specifically in the strong updraft zones
and the seed particles were also lifted inside the cloud and more cloud droplets were noted both in
the observations and simulations. Earlier, during the Seeded and Natural Orographic Wintertime
Clouds: The Idaho Experiment (SNOWIE) (Xue et al., 2022) noted seeding plumes dispersed within
orographic clouds in more than 1 hour along the slanted downwind direction.

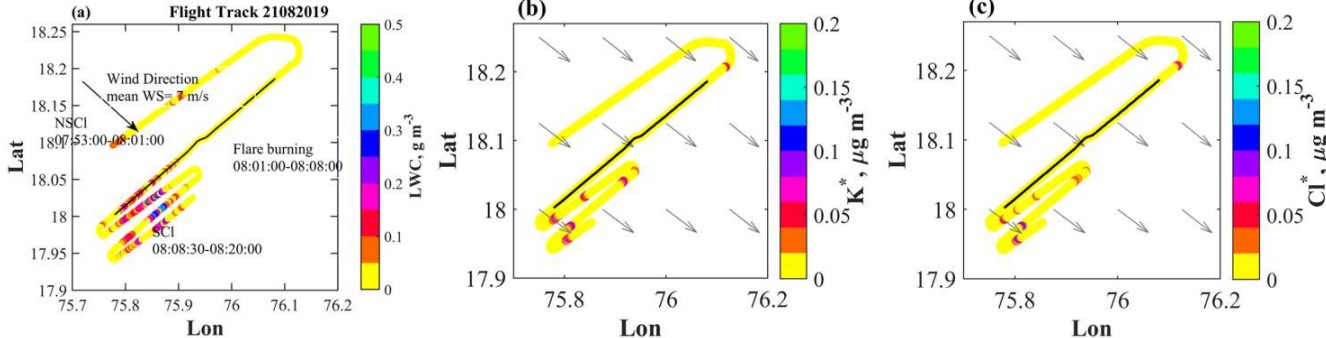

**Figure 5**. (a) The flight path during the seeding experiment on 21 August 2019 color coded by LWC
at 1 Hz resolution. Periods during which cloud measurements were made for non-seeded clouds
(NSCl) and seeded clouds (SCl) are annotated.  Mass concentrations of (b) K$^*$ and (c) Cl$^*$ during the
seeding experiment are shown along the flight track. The ambient wind fields shown as arrow
obtained from https://cds.climate.copernicus.eu/   (0.25 ºX0.25 º), which are resampled to 0.125 º X
0.125º. A small area of elevated K and Cl, prior to the flare burning is noted. This was measured
outside the cloudy region as suggested by the LWC values and it might be appeared probably due to
other unknown sources.

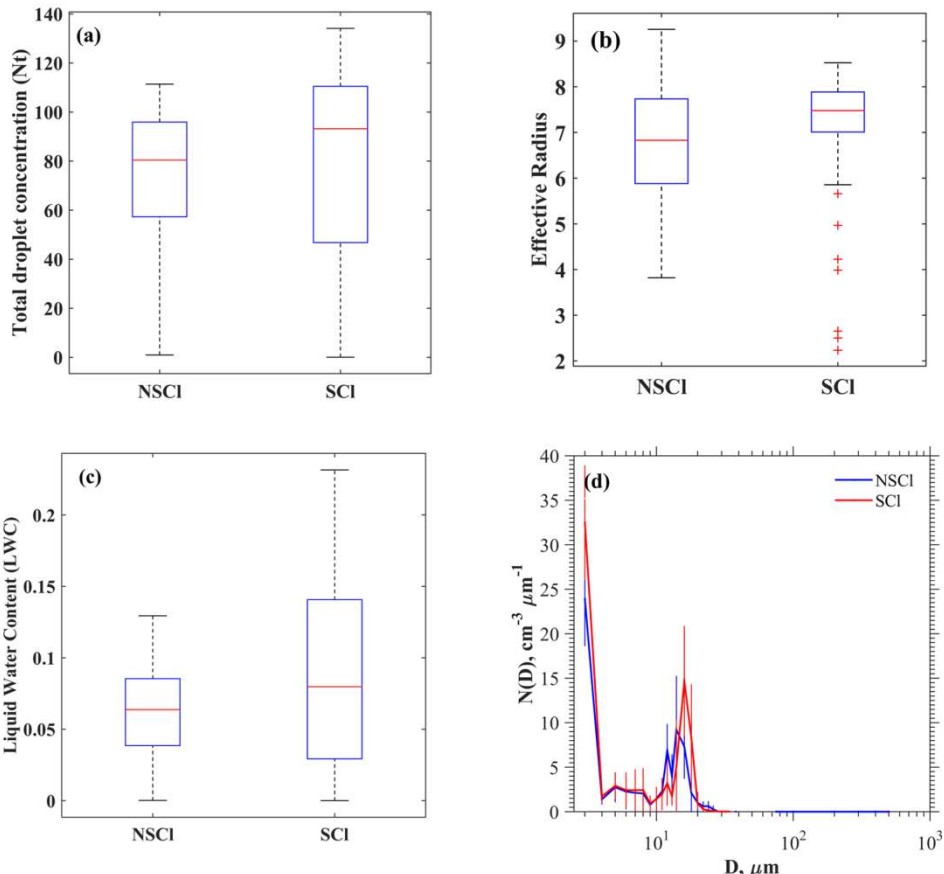

**Figure 6**: Box plots of (a) total droplet concentrations, (b) Effective radius, (c) LWC are shown for NSCl and SCl. (d) Mean cloud DSDs with standard deviations (vertical bars) are depicted indicating the variability. The selected DSDs fall within the criteria of $0.75 < \text{LWC}/\text{LWC}_{\text{max}} < 1$.

Stratus cloud passes were selected for study based on two criteria: a cloud pass duration greater or equal to 5 seconds and $N_t>10$ cm$^{-3}$. Two NSCl cloud passes made during 7:53:00-7:53:31 UTC and 7:55:17-7:55:41 UTC were chosen for the analysis. After the flares had dispersed, three passes during 08:08:37-08:08:45 UTC, 8:09:42-8:09:53 UTC, and 8:09:59-08:10:39 UTC were selected based on the elevated levels of detection of K and Cl ( see Fig. 5b and 5c). Box plots of $N_t$, $r_e$ and LWC are displayed for NSCl and SCl in Figs. 6a, b and c, respectively. It is worth noting that the SCl cases exhibit greater median values for these three parameters. The properties of DSDs along

the cloud pass are shown in Supplementary Figs. S1 and S2. The DSD properties and mass
concentrations of K* and Cl* are provided in Table 3. Increased droplet concentrations in the
smallest size bin are noted after a few minutes from the seeding time while drizzle drops were not
observed in the SCl. Comparsions are made for mean SCl-DSD and NSCl-DSD in the range
$0.75 < LWC/LWC_{max} < 1$, as illustrated in Fig. 6d. An increase of N(D) at D ≈ 3 μm and in the size
range 13 < D < 20 μm are noted in the SCl, while N(D) decreased in the size range 4 < D < 13 μm.
The increase in the smallest cloud droplets may be due to freshly nucleated aerosols, likely due to
the activation of seeding materials. The increase in the mid-size droplet concentrations could be due
to the activation of coarse mode aerosols and subsequent diffusional growth. Since drizzle drops
were not formed, it may suggest that hygroscopic seeding in stratus cloud with low LWC value e.g.
$< 0.5$ g m$^{-3}$ may not yield a significant positive seeding effect for the production of drizzle.

**3.2.2   Case ii: 23 August 2019**.
Fig. 7a depicts the flight patterns for the case on 23 August 2019. This seeding event is selected for
evaluation because (i) The SCl and NSCl convective clouds were isolated and in the growing and
non-precipitating stages, (ii) the cloud top was below freezing level (5 km) therefore ideal for
studying warm rain microphysics, (iii) The SCl and NSCl were formed within the same area (20 km
x 20 km) and lastly, (iv) both the SCl and NSCl grew to similar cloud top altitudes (≈4 km),
therefore roughly at similar growth stages. These conditions made this case suitable for evaluating
the seeding effect on warm rain. The cloud base height over the observational area was nearly 1.80
km. Northwesterly winds (mean wind speed of 12 ms$^{-1}$) prevailed in the boundary layer at 1.30 km
(850 mb). Before the dispersal of flare materials  at cloud base, the cloud microphysical properties of
NSCl were measured from 7:49 to 8:06 UTC by step-wise multiple cloud penetrations from the top
($\approx$ 3.90 km) to near the cloud base ($\approx$ 1.80 km). A maximum updraft of 4.40 ms$^{-1}$ was observed at
the cloud base. After completion of NSCl measurements, the aircraft then circled below the cloud
base and burned four hygroscopic flares (two on each wing) in the updrafts during 8:08-8:12 UTC,
followed by several step-wise cloud penetrations at nearly 1000 ft intervals, from near the cloud base
to cloud-top during the period 8:14-8:28 UTC.

404         The profiles of $N_t$ and $r_e$ *w.r.t.* the D*s are shown in Fig. 7(b,c). The mass concentrations of

K* and Cl* corresponding to $N_t$ and $r_e$, respectively, are also indicated. The statistical properties of
the DSD parameters are presented in Table 3. The variations of DSDs along the cloud transects,
values of $r_e$, drizzle concentration, LWC, and W are shown in the supplementary material's Figs. S3-
4. Note that the SCl and NSCl were not identical due to the natural variability discussed previously,
with this background the following  observations are noted:


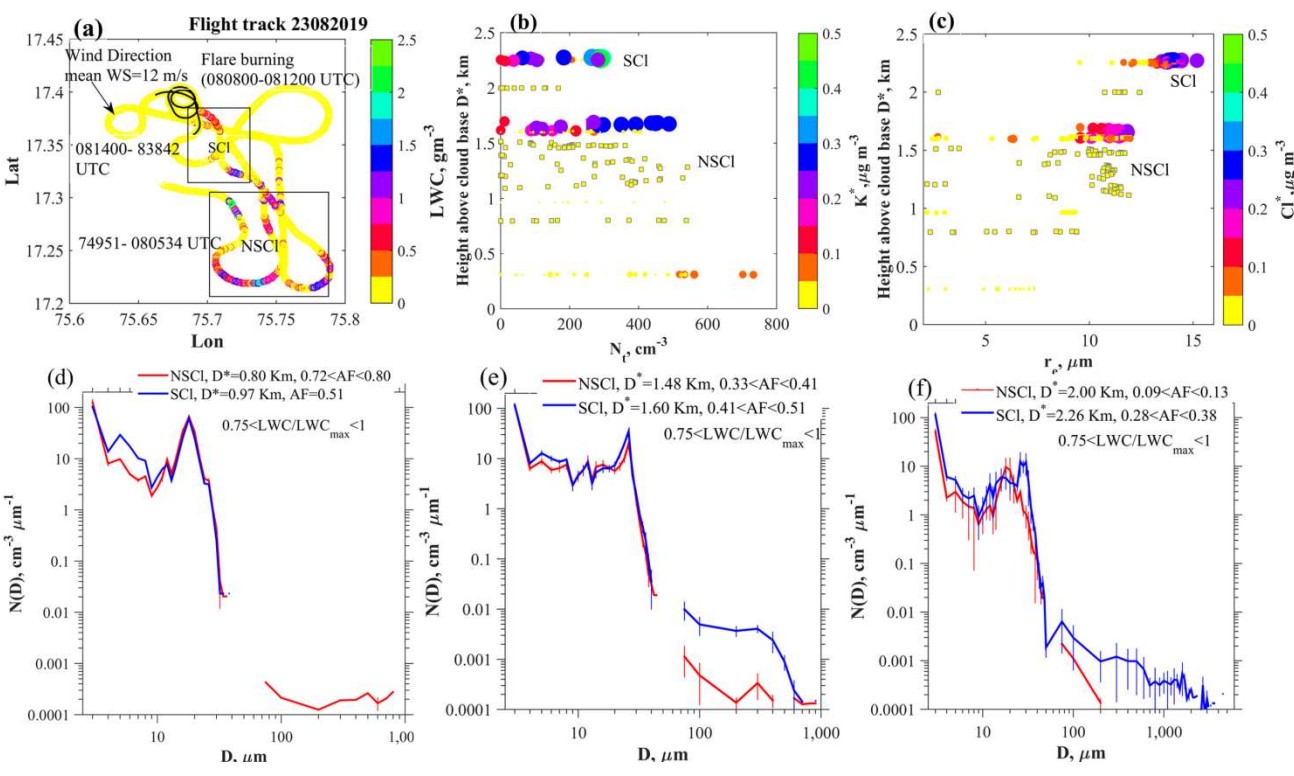


**Figure 7**. (a) Flight track during the seeding experiment on 23 August 2019. The flight track during

the flare burning period is overlaid with black color. The areas of seeded cloud (SCl) and non-seeded

cloud (NSCl) are indicated on the figure panels. The arrow indicates the wind direction near the

cloud base height of 1.80 km. The color bar indicates the liquid water content (LWC, $gm^{-3}$) of

clouds. Profiles of (b) $N_t$, ($cm^{-3}$) and (c) $r_e$, (µm) *w.r.t.* height above cloud base, D* (km) are shown.

The parameters are indicated in the color bars with the mass densities of K* and Cl*, (µg $m^{-3}$). The

squares with black edges indicate NSCl, while filled circles indicate SCl. The sizes of the symbols

increase with increasing mass of the chemical components. Mean cloud drop size distributions with

standard deviations indicated by the error bars of slightly diluted clouds (0.75<LWC/$LWC_{max}$ <1) at

various D* (km), for NSCl and SCl, (d), (e) and (f).


Table 3.
Cloud properties of Non-Seeded Cloud (NSCl) and Seeded Cloud (SCl) along the cloud transect are
shown. Vertical distance above the cloud base (D*, km), Mean values and standard deviation of total
droplet concentration $N_t$, (cm$^{-3}$) in the diameter range 2-50 µm, maximum droplet concentration
($N_{tmax}$, cm$^{-3}$), mean effective radius ($r_e$, µm), liquid water content (LWC, gm$^{-3}$), Maximum LWC
(LWC$_{max}$), maximum adiabatic fraction (AF$_{mx}$ = LWC$_{max}$/LWC$_{ad}$), where LWC$_{ad}$ is the adiabatic
LWC calculated from a parcel model. AF$_{mx}$ for layer clouds on 21082019 is not calculated. The
mean of small droplet concentration (D<11 µm) and the maximum of small droplet concentration,
and drizzle concentration (DrizzleCon, (cm$^{-3}$) are also shown. Concentrations of K* and Cl* in µg m$^{-3}$
during NSCl and SCl observations are indicated. Due to limited field calibrations, the
concentrations presented here are nitrate equivalent. Below Detection Limit (BDL) data are
indicated.

| Case | D* (km) | N$_{tmn}$ ±SD (cm$^{-3}$) | N$_{tmax}$ (cm$^{-3}$) | r$_e$ ±SD (µm) | LWC ±SD (gm$^{-3}$) | LWC$_{max}$ (gm$^{-3}$) | AF$_{mx}$ | N$_{tmn}$, [N$_{tmx}$] (D<11µm) | DrizCon ±SD (cm$^{-3}$) | Mean K* ±SD [K*$_{Max}$] mg m$^{-3}$ | Mean Cl* ±SD [Cl*max] mg m$^{-3}$ |
|---|---|---|---|---|---|---|---|---|---|---|---|
| 2108-NSCl | 0.35 | 73±23 | 105 | 7.28±1.22 | 0.07±0.03 | 0.13 | - | 46±20[89] | 0 | BDL | BDL |
| 2108-NSCl | 0.40 | 73±35 | 111 | 5.93±1.03 | 0.05±0.03 | 0.13 | - | 39±20 [77] | 0.004±0.02 | BDL | BDL |
| 2108-SCl | 0.07 | 47±40 | 108 | 7±1.50 | 0.05±0.05 | 0.13 | - | 21±16 [49] | 0±0 | 0.0024±0.001 [0.004] | 0.003±0.0005 [0.004] |
| 2108-SCl | 0.08 | 62±40 | 111 | 6.05±1 | 0.05±0.04 | 0.10 | - | 42±28 [80] | 0±0 | 0.06±0.03 [0.09] | 0.02±0.02 [0.06] |
| 2108-SCl | 0.08 | 92±35 | 134 | 7.54±0.86 | 0.11±0.06 | 0.23 | - | 44±17 [79] | 0±0 | 0.003±0.004 [0.02] | 0.0005±0.0003 [0.001] |
| 2308-NSCl | 1.99 | 65±60 | 167 | 10.72±2.86 | 0.19±0.17 | 0.48 | 0.13 | 30±27 [68] | 0±0 | BDL | BDL |
| 2308-NSCl | 1.48 | 177±104 | 360 | 9.70±2.42 | 0.42±0.34 | 1.11 | 0.41 | 101±57 [185] | 0.01±0.01 | BDL | BDL |
| 2308-NSCl | 1.33 | 254±173 | 541 | 10.26±1.31 | 0.69±0.48 | 1.57 | 0.61 | 121±84 [262] | 0.01±0.01 | BDL | BDL |
| 2308-NSCl | 1.16 | 254±184 | 528 | 9.40±3.22 | 0.80±0.66 | 2.00 | 0.88 | 116±75 [210] | 0.31±2.65 | BDL | BDL |
| 2308-NSCl | 0.80 | 208±198 | 538 | 6.57±2.60 | 0.32±0.44 | 1.22 | 0.80 | 107±84 [221] | 0.05±0.04 | 0.001±0.0005 [0.001] | BDL |
| 2308-SCl | 0.31 | 402±194 | 733 | 6.74±0.84 | 0.42±0.22 | 0.69 | 0.92 | 144±69 [323] | 0±0 | 0.03±0.22[0.08] | 0.014±0.01 [0.02] |
| 2308-SCl | 0.31 | 236±192 | 482 | 5.90±1.64 | 0.23±0.20 | 0.54 | 0.72 | 90±67 [169] | 0±0 | 0.004±0.003 [0.01] | 0.0005±0.0002 [0.0008] |
| 2308-SCl | 0.96 | 186±158 | 477 | 7.30±3.01 | 0.35±0.31 | 0.97 | 0.51 | 81±71 [196] | 0.002±0.007 | 0.005±0.001 [0.008] | 0.011±0.003 [0.015] |
| 2308-SCl | 1.64 | 200±139 | 488 | 10.41±1.50 | 0.62±0.51 | 1.74 | 0.57 | 83±53 [198] | 0.53±0.50 | 0.17± 0.10 [0.29] | 0.12± 0.08 [0.21] |
| 2308-SCl | 1.60 | 162±120 | 332 | 9.70±3.00 | 0.50±0.38 | 1.04 | 0.34 | 71±54 [157] | 0±0 | 0.003±0.001 [0.005] | 0.003± 0.001 [0.004] |
| 2308-SCl | 1.60 | 184±139 | 404 | 9.50±2.82 | 0.57±0.58 | 1.55 | 0.51 | 95±63 [183] | 0.41±0.43 | 0.01± 0.01 [0.02] | 0.023±0.02 [0.08] |
| 2308-SCl | 2.26 | 175±107 | 320 | 13.10±1.14 | 0.80±0.50 | 1.49 | 0.38 | 83±51 [155] | 0.43±0.52 | 0.18±0.12 [0.40] | 0.11±0.10 [0.28] |
| 2408-NSCl | 0.21 | 92±92 | 244 | 5.55±1.76 | 0.06±0.06 | 0.18 | 0.31 | 56±59 [147] | 0±0 | 0.0008±0.0003 [0.001] | 0.002±0.002 [0.005] |
| 2408-SCl | 0.20 | 159±153 | 413 | 5.57±1.76 | 0.14±0.15 | 0.41 | 0.70 | 65±57 [157] | 0±0 | 0.002±0.001 [0.003] | 0.001±0.001 [0.002] |
| 2408-SCl | 0.20 | 161±189 | 649 | 5.91±2.06 | 0.16±0.18 | 0.56 | 0.96 | 70±88 [321] | 0±0 | 0.01±0.01 [0.02] | 0.004±0.003 [0.01] |
| 2408-SCl | 0.20 | 300±171 | 603 | 6.58±1.30 | 0.32±0.19 | 0.54 | 0.93 | 111±72 [347] | 0±0 | 0.02±0.01 [0.05] | 0.01±0.01 [0.02] |


(i)       At nearly $D^* = 0.96$ km, smaller mean concentrations of $N_t$ ($186\pm158$ cm$^{-3}$) are noted for
SCl compared to the NSCl ($N_t = 208\pm198$ cm$^{-3}$) cloud pass at $D^* = 0.80$ km. At these two nearly
similar levels, the mean $r_e$ values for the SCl case ($r_e = 7.30\pm3.01$ μm) were greater than those for the
NSCl case ($r_e = 6.57\pm2.60$ μm). At greater $D^*$ of 1.60 km ($r_e = 9.50\pm2.82$ μm) and 2.26 km
($r_e=13.10\pm1.14$ μm), drizzle drops (see Table 3) were noted in the SCl cases. This may indicate
active CC process in the SCl case. The mean DSDs are shown in Fig. 7(d,e) selected considering the
criteria $0.75< \text{LWC/LWC}_{max}< 1$ of the cloud transects. The corresponding AF values indicated on
the panels suggest active entrainment and mixing processes in these clouds. The production of
drizzle in some of the clouds may also lower the AF values which means that the dilution rate is not
accurate in such clouds. The seeding effect may give rise to the initial production of drizzle particles,
which were seen within the tail of the DSDs. Hence, the tail effect of the seeding particles appears to
be active. Note that since the cloud passes were made in the developing stage of the cloud, these
drizzle drops were formed spontaneously, not falling from the cloud tops because their terminal
velocities are less than the updraft velocities. The broadening of the DSDs will serve to further
increase the efficiency of the CC process (Andreae, et al, 2004; Rosenfeld et al., 2008; Rosenfeld et
al., 1994; Freud et al., 2012; Konwar et al., 2012) leading to the production of drizzle drops at higher
$D^*$s. Also, stronger updrafts ($\approx 5$ ms$^{-1}$) were observed in SCl (see Fig. S4n), which helped in the
growth of larger-sized droplets.
The formation of drizzle drops (D>100 μm) in the SCl was noted (Fig. 7(e,f) and Fig. S4) while no
significant drizzle concentrations were noticed for NSCl (Fig. S3). The difference in drizzle
concentration suggests that the flare particles modulate the mid-size cloud droplets (D $\approx$ 14 μm) that
grow further by diffusion process. As the drizzle drops fall under the influence of gravity, stronger
downdrafts are most likely due to the cooling by evaporation (see Fig. S4n). Moreover, small
droplets of D≤11 μm were observed at high altitudes for both clouds (Table 3). The scatter plots
between $r_e$-K$^*$ and $r_e$-Cl$^*$ are shown in Fig. S5. The prevailing dynamical conditions e.g., vertical
velocity are also indicated. It is found that the larger sized droplets (greater $r_e$ values) are associated
with the larger mass concentrations of K$^*$ and Cl$^*$, in the SCl. In both the updrafts and downdrafts,
all these chemical species were present. Having found the seeding tracers Cl$^*$ and K$^*$ at different
altitudes, it may be emphasized that the modification of cloud properties occurs due to the dispersal
of seeding particles through the cloud base. Seeding particles were present at deeper D*s as the
cloud droplets were transported through updrafts and re-circulated as the cloud developed (Khain et
al., 2013).
It is important to note that the differences in cloud microphysical properties observed between the
seeded and unseeded clouds could be a result of natural variability, and more data are needed to
arrive at a  statistically significant result. However, given that these differences were accompanied
by statistically different concentrations of chemical composition in the cloud droplet residues in the
same environmental conditions, the evidence is compelling that seed material has a) transported to
altitudes above the cloud base where they were released and b) these aerosol particles have
influenced cloud microphysical processes.

### 3.2.3  Case iii: 24 August 2019.
The third cloud seeding case was carried out on an isolated convective cloud. The flight path is
shown in Fig. 8a. South-westerly winds with a mean speed of 9 m s$^{-1}$ were noted near the cloud base
at 2.1 km with a maximum updraft of 8 m s$^{-1}$. One cloud pass before the flare dispersal was made
from 08:55-08:59 UTC above the cloud base at  ≈ 2.3 km. Three downwind cloud passes during
09:05-09:07 UTC were made at $\approx$ 2.3 km after the flares were burned. The variations of $N_t$, and $r_e$
*w.r.t.* D* are shown in Figs. 8b,c. Increased mass concentrations of $K^*$ and $Cl^*$ are noted in SCl cases
that identify the seeded clouds. The DSD properties of the clouds are shown in supplementary Fig.
S6 & S7  and their parameters are indicated in Table 3. The mean DSDs (Fig. 8d) indicate increased
droplet concentration in the small and mid-drop diameter ranges. Note that the AF values indicated
strong dilution in the NSCl DSDs, which may also impact the observed differences in the droplet
number densities. No marginal increment in $r_e$ values was observed in the SCl.  Another aspect to
consider here is the effect of strong updraft of 8 m s$^{-1}$. Using the Twomey (1959) equation the
maximum droplet concentration formed in an updraft (W) can be expressed in terms of W and CCN-
SS spectra, i.e. $N_{CCN}=C\,SS^k$ i.e. (Roger and Yau,  1989),
$$N \approx 0.88\, C^{2/(k+2)}\left[7\,X\,10^{-2}\,W^{3/2}\right]^{k/(k+2)} \qquad (4)$$

Here, W is in cm s$^{-1}$, $N_{CCN}$= 799 $SS^{0.43}$, which is obtained from the CCN counter (Roberts and
Nenes, 2005; Nenes et al., 2001 and reference therein) operated in the research aircraft. During the
cloud passes, maximum updrafts of W= 2.89 m s$^{-1}$, 1.00 ms$^{-1}$ and 1.91 m s$^{-1}$ were obtained. These
values suggest that droplets formed in these updrafts could be 593 cm$^{-3}$, 448 cm$^{-3}$ and 531 cm$^{-3}$,
respectively. If we use the maximum updraft speed of 8 ms$^{-1}$ measured below cloud base, the droplet
concentrations formed in this updraft could be as high as 777 cm$^{-3}$. In this scenario, the
supersaturation could be greater than 1%, which can activate small-sized CCN. Therefore, the
presence of strong updrafts that yield high SS could be one reason for the increasing $N_t$ in the seeded
clouds; while dry air mixing in the NSCl cases could be another reason for the smaller concentration
of $N_t$. These processes may be attributed for the change in LWC values in the SCl cases.


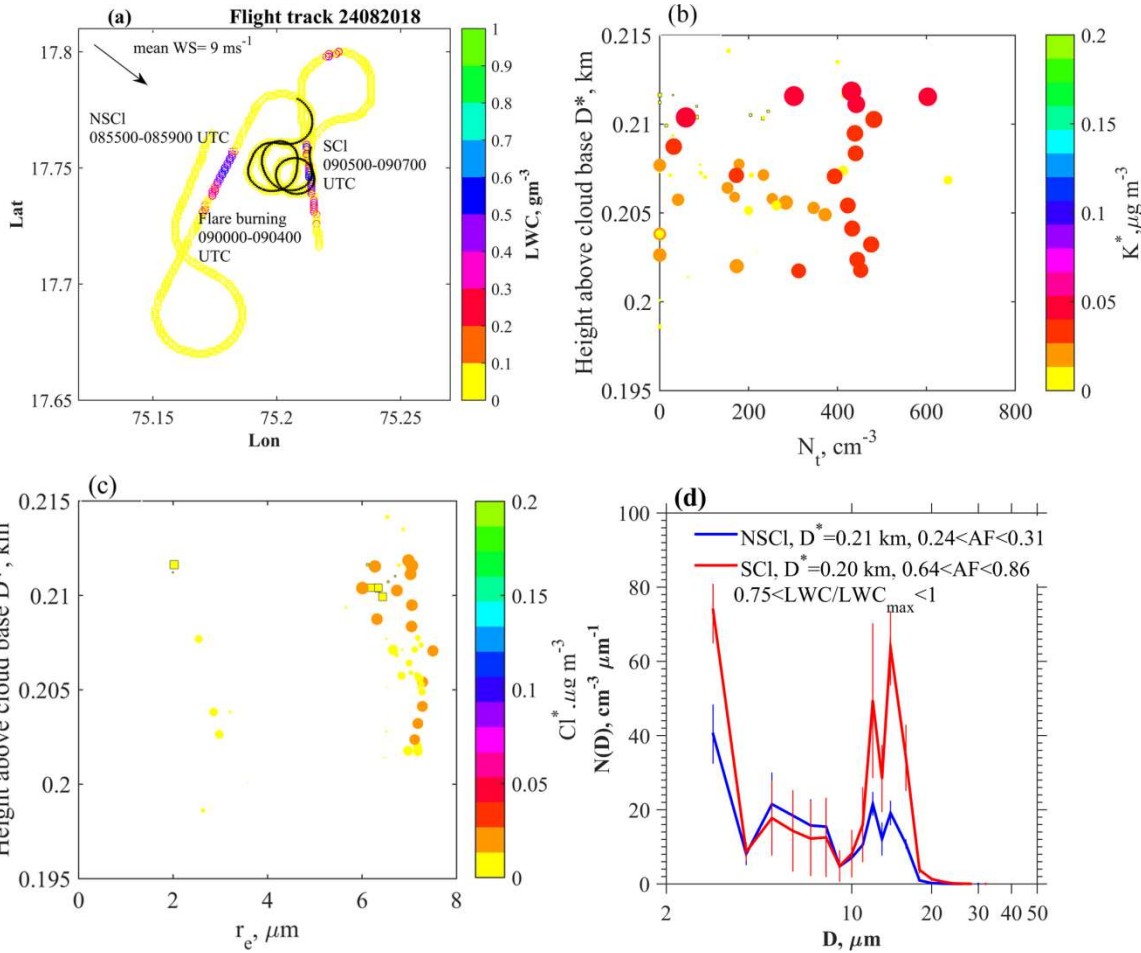


**Figure 8**. (a) Flight path during the seeding experiment on 24 August 2019. Periods during which
cloud measurements were made for NSCl and SCl are indicated. The black line indicates the flare
burning. Profiles of (b) $N_t$, and (c) $r_e$, *w.r.t.* D* (km). The parameters are indicated with the mass
concentrations of K*, (µg m$^{-3}$), and Cl* (µg m$^{-3}$). (d) Mean DSDs with standard deviations indicated
by the vertical bars, of clouds (0.75<LWC/LWC$_{max}$ <1) above the cloud base, for NSCl and SCl. The
adiabatic LWC fractions corresponding to the DSDs are also indicated.



## 4. Summary and conclusions:

The successful identification of seeded cloud hydrometeors, and the tracing back to their seeding origins in cloud seeding experiments has been an outstanding challenge for cloud seeding operations. The unequivocal identification of seeding material within clouds was the primary difficulty in such experiments. During the CAIPEEX 2019 seeding experiments conducted in India, we measured cloud microphysical properties and traced the seeding material with an mAMS behind a CVI in convective and stratus clouds.

In our experiments, the mAMS identified an enhancement of both K and Cl mass concentrations, most likely from the oxidizing agent (KClO) and seed material ($CaCl_2$). In stratus and convective clouds, such enhanced concentrations of refractory K and Cl should be considered as a seeding signature.

Enhanced small-sized droplet concentrations that were measured near the cloud base of convective clouds and in a warm stratus layer are noted. This result indicates that during the monsoon season with an available moisture supply, even the small-sized CCN present in the seed material could be activated into cloud droplets. The presence of strong updrafts near the cloud base of isolated convective clouds could also play a major role in the activation of small-sized CCN to cloud droplets. These strong updrafts would yield high supersaturation values, thus activating small-sized CCN. The impact of strong updrafts on the activation of cloud droplets, especially when seeding agents are dispersed below the cloud base, requires more focused attention and study.

In the case of a convective cloud, clear differences in the cloud microphysical properties of SCl compared to NSCl are noted. The flare materials released below the cloud base were lifted to a height of 2.25 km above the cloud base. In the lower part of the SCl larger droplet concentrations

were noted. The SCl also had a larger $r_e$ than the NSCl at similar heights above the cloud base. The
seeded clouds contained more drizzle drops, suggesting that they reached the threshold for warm
rain initiation at a lower distance from the cloud base than the non-seeded clouds. These results from
the limited sample indicate the plausible tail effect of the largest particles in the flares, initiating
large cloud drops and drizzle. Though this case study indicate the importance tails effect; conclusive
evidence would require much more data.
Whether competition or the tail effect is important in a successful cloud experiment remains to be
examined, as the prevailing dynamical conditions can play a significant role in controlling the cloud
microphysical processes. These complexities need to be addressed with more experiments using
mAMS.
This study identifies a novel methodology to simultaneously track and measure the cloud seeding
signatures and to assess how the seeding alters the microphysical properties of clouds leading to
raindrop formation. The utilization of an mAMS in cloud seeding experiments together with a CVI
allows for identifying the seeded cloud parcels of interest, leading to a better understanding of the
effects on the microphysical properties of the cloud. Although these measurements of flare material
in seeded clouds are associated with changes in physical properties, the data set is too limited to
unequivocally assert that this methodology will always be successful. Future studies with a much
larger data set will provide more statistical evidence linking seed aerosol and increases in
precipitation.
**Acknowledgment:** Indian Institute of Tropical Meteorology, Pune and the CAIPEEX project are
funded by the Ministry of Earth Sciences, Govt. of India. We thank Director, IITM for continuous
supports. The authors are grateful to the team members, the ground staff, V. Ruge and S. Patil of
M/S Tesscorn AeroFluid, Inc., and the pilots for their dedicated efforts in conducting the project.
The authors are grateful to the Editor and two anonymous reviewers for their insightful suggestions
that helped improve the manuscript.

**Data availability**

mAMS and Cloud data are available at:
https://iitmcloud.tropmet.res.in/index.php/apps/files/?dir=/&fileid=59847#

**Author contributions**

TP and DW designed the mAMS experiment; MK, BW and ECF prepared the initial draft;  KH,
MK, BW, ECF, SC, SB, NM, MV, SJ and TP participated in the aircraft experiment; DB, TP, DW,
DA, PM, MK, BW, ECF, MV, SC,SB and SAD reviewed the manuscript. All authors agree with the
final version of the manuscript.

**Competing interests**

The contact author has declared that none of the authors has any competing interests.

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
