# Peer review of "Identifying the seeding signature in cloud particles from hydrometeor"

_Atmospheric Measurement Techniques, 2023_

## Referee Comment (RC2)

Review of 'Identifying the seeding signature in cloud particles from hydrometeor residuals'

Konwar et al.

AMT-2023-171

*Summary*
In this manuscript, the authors present airborne measurements from a mini-Aerosol Mass Spectrometer (mAMS) to identify regions in cloud that may have been influenced by cloud seeding. Following sampling using a counter flow virtual impactor (CVI) that removes hydrometeors and other aerosol with aerodynamic diameters smaller than a certain size (~7 um in this study), cloud droplets are evaporated and the residual aerosol are tested for the presence of chemical species, including potassium (K) and Chlorine (Cl) that are otherwise not present (or not detectable in natural form) within the natural cloud environment. The authors (correctly) state that unambiguously identifying cloudy regions in which seeding material is contained within hydrometeors remains elusive; and developing a method to identify such regions is a key step to evaluating hypotheses associated with how cloud seeding might impact cloud and precipitation evolution. In this manuscript, the authors present evidence that they are able to do just that, and thus I believe this work makes an important contribution and is therefore worthy of publication. However, there are numerous points (some major) that I feel the authors need to address before publication.

In the first case study (case I, on Aug 21); the authors obtain measurements upwind in a stratus cloud and follow that with a release of seeding material a bit downwind, which is again followed with additional passes downwind to measure the cloud after seeding. The data shown in Figure 5 (b and c) indicate that most of the measurements downwind of the seeding release show elevated K and Cl. That is somewhat counter to what I would expect. The seeding material should disperse downwind in a slantwise pattern…similar to that shown in Tessendorf et al (2019; BAMS) and other references coming out of SNOWIE. Thus, when flying the 3 passes downwind in this case, I would expect the aircraft to fly through a brief line (region) containing seeding material surround by most of those legs being outside of regions containing seeding material. The authors could develop a simple advection model utilizing the mean wind speed and the time of the release along the seeding leg to estimate how the seeding line advects downwind. That would then provide an estimate of where to (and where not to) expect the presence of seeding material in the cloud.

That the authors are able to detect the presence of K and Cl is exciting, however, I think the onus is on them to prove that it is related specifically to the region that contains seeding material. My gut instinct is that it indeed is identifying those regions, but I am still a bit concerned based on the measurements provided in figure 5 as I describe in the above paragraph, that the method 'over-identifies' regions containing seeding material. I also wonder, since the same aircraft is used for releasing seeing material and making measurments; is there any potential for contamination? Were any experiments conducted where NSCl was re-sampled upwind following the release of seeding material?

*Major Comments*

1. Introduction—I think the introduction could be stronger. For example, in the second and third paragraphs the authors discuss the use of tracer materials to identify how researcher might understand the transport of seeding material through clouds. They describe two technologies previously used (chaff and chemical tracer such as SF6); while describing downsides of both, I think they miss the main point that *neither of those methods* would allow investigators to determine if seeding material actually makes it into cloud hydrometeors. This is touched on around line 85 ('can changes is cloud microphysical processing be linked to seeding materials'); in order for this to happen, the seeding material must actually interact with hydrometeors. The method developed in this manuscript does exactly that, by sampling residuals from cloud droplets. The author's should highlight this

2. Lines 131 – 139; and Table 1: Measurements of Cloud properties. The authors need some additional discussion on uncertainties in their measurements. These should come from published literature and simply referencing the manufacturer website is not sufficient. For example, the CDP provides a measurement/estimate of the size distribution, but there are uncertainties associated both with counting (though likely small) and sizing (which could be significant in some size bins)…these errors/uncertainties can lead to further (and much larger) uncertainties in integrated quantities such as liquid water content. The authors use LWC as a threshold in this study to identify regions less likely to be impacted by entrainment/mixing; thus having a discussion of potential errors in this measurement is important.

3. It appears that in the three cases presented, the authors knew well the cloud base conditions. Lines 144-5; the authors describe a threshold method to identify parcels that are either not or only slightly diluted. Why use LWC_max rather than LWC_ad? A threshold of 0.75 * LWC_max could be a highly diluted if the maximum LWC during that penetration is significantly less than the adiabatic value. Using adiabatic can be a direct estimate of 'how diluted' a parcel may be.

4. Specific for Seed Case i: Fig 5, Table 3, and paragraph contained in lines 329-343 – Several questions come up when I look at the presentation of this case: How are the in-cloud times selected for which dots are represented in figs 5b and 5c? I don't think I'd call these 'profiles' in fig5 b & c because all thee downwind (SCl) passes are at one level and the the NSCl pass is at a slightly higher level. I understand wanting a consistency in figure style between this case and the other two; but presenting this case in this style doesn't make sense to me. The data in table 3 indicates that one of the NSCl cases in this case had a LWC of 0.003 +/- 0.003 – this seems marginal as a cloud to me; less than 0.01 LWC is below what most would consider in cloud. It seems odd to include data such as this as a comparison of cloud? LWCs are generally higher after seeding (although the measurements are lower in cloud…why? Seeding should not result in increased condensed water, but rather just changes in how that condensed water is distributed across hydrometeor distributions. On line 340 the authors suggest that increase in midsized droplet concentration may be due to collection, but this does not make sense…these droplets are <20 um; collection is extremely inefficient at these.

5. For Seed Case ii: In Fig 6 b and c – Need a better way to differentiate between which dots represent NSCl and which represent SCl – perhaps shape? One could be square—the other circle? Line 392 – the authors claim that after seeding Nt increased at lower altitudes—I disagree. The lowest altitude above cloud base sampled before seeding was 0.8 km and after seeding was 0.3 km – these two cannot be compared---the so seed is more than twice the distance above cloud base. Further, in the after seeding case, a pass was made at 0.96 km, and that contained lower concentrations than the no seed at 0.8 km. The authors need more rigor in their data analysis before making such claims. On line 398 the authors claim that large standard deviations in re are likely the result of entrainment/mixing with dry air….I agree; but then they need to be more rigorous in trying to eliminate regions which are influenced by mixing in their analysis. This is a possible indicator that their threshold of 0.75*LWC_max is not sufficient to do accomplish this. On line 412 the authors suggest that because drizzle is encountered after seeding and not before that the production of drizzle can be attributed to seeding…while this is one possibility, there are many other possibilities that could account for that. Different stages of lifecycle, pre-conditioning of the environement, etc are all possibilities. Studies of cumulus continually show that drizzle may form at the same levels in a cumulus cluster that indicated no drizzle development several minutes earlier.

6. For Seed Case iii: Why is the LWC 3-5 times higher after seeding than before, when observations were made at the same level above cloud base? One possibility is that these difference are due solely to difference in amount of dry air mixed into parcels that were sampled, and any differences in microphysics may be attributed to that rather than to seeding?

*Specific Comments*
1. Abstract, Line 31 – ….attempts to link precipitation enhancement….remained inclusive? Not sure what is meant by this, could it be the authors mean that attempts remained elusive?
2. Abstract, Line 37 -- ….cloud droplets underwent a drying process, … I think it would be more accurate to state that droplets were evaportated.
3. Line 67 – '…then these traces are tried to measure higher in the cloud.' Wording here is incorrect.
4. Line 91 – '…to identify seeding material in the residual cloud droplets.' The seeding material is identified within 'cloud droplet residuals'; i.e. the aerosol that remains *after* evaporation of the cloud droplets.
5. Line 92 – '….hypothesis relies on a chain of microphysical mechanisms.' The wording here is not correct; there is no such thing as a chain of mechanisms. I think the authors are referring to a chain of events with each event being a specific microphysical process.
6. Line 99 – 'CCN do..' (not does, CCN is plural)
7. Line 107 – replace 'seed' with 'seeding agent' and delete the word 'the' before broadening.

8. Sentence on lines 111-113 – I read this sentence in relation to the evolution of microphysical processes resulting from hygroscopic seeding (previous sentence); however, this has been accomplished in glaciogenic seeding experiments as the authors reference French et al. (2018). It seems to me that the processes are *more* dynamic and complex in cold clouds, yet that linkage has been made.

9. Lines 127-129 – isn't all the data presented in this study from 'warm' cloud seeding? Why reference flare racks for 'cold' cloud seeding. And, by 'warm' and 'cold' are the authors referring to hygroscopic vs glaciogenic seeding? If so, they should be referred to as such.

10. Use of abbreviations and nomenclature for height of measurements in cloud— introduced on Lines 146 & 7 – The authors use the nomenclature 'cloud depth' (abbreviated CD) to describe the heights (above cloud base) for measurements throughout the manuscript. I find this confusing as 'cloud depth' normally refers to a measurement of the total depth (or thickness) of a cloud. I suggest the authors should change all instances in text, figures, and tables from 'cloud depth' to 'height above cloud base'.

11. Line 161 – Delete 'As mentioned earlier'

12. Line 165 & 6 "…the CVI inlet segregates and samples cloud elements." Replace that with: 'the CVI inlet cloud hydrometeors with aerodynamic diameters larger that a certain size depending the rate of the counter flow.

13. Line 168 – 'larger than (D > 7 um)' how was this verified for the setup in this experiment? The cutoff size is dependent not only on flow rate, but also aircraft speed, possibly mounting location on the aircraft, altitude, etc. Verifying this (somehow) seems to be a critical thing in this experiment.

14. Line 172 – I cannot access the Golderger et al. (2020) reference.

15. Lines 230 & 231  and Figure 3—Is 'start' and 'stop' the times of 'Pass Begin' and 'Pass End' shown in the figure? If so, be consistent in labeling. Label 'maximum total mass time' in figure 3.

16. Lines 253 through 260 – The authors repeat items that are already discussed earlier and restate the same things multiple times in lines in this paragraph. I think the authors should rewrite this paragraph to make it more concise and less repetitive.

17. Line 285 and 286 reword to: …'with three cloud passes of the same cloud before and three passes after seeding are shown in Fig 4.'

18. Line 288 – change 'above' to 'compared to'

19. Line 290 – the wording seem awkward here…maybe change 'consist of' to 'containing'

20. Line 291 & 2 – I do not see increased SO4 concentrations in the seeded clouds (perhaps at mid level, but certainly not at the highest level…

21. Line 307 – 'several cloud passes….were made….before dispersal of seeding material.' From Figure 5a, it looks like one pass was made.

22. Line 311-314 – The AIMMS 20 does not measure whether in cloud, I assume the authors use some threshold on the CDP measurements? Possibly a threshold on the LWC derived from the integrated size distribution?

23. Line 326 – sentence that begins 'Discussions on cloud probes…' ---delete this.

---

## Author Comment (AC1)

Identifying the seeding signature in cloud particles from hydrometeor residuals
Konwar et al.

**Responses to Reviewer#2**

Review of 'Identifying the seeding signature in cloud particles from hydrometeor residuals' Konwar et al. AMT-2023-171

Summary: In this manuscript, the authors present airborne measurements from a mini-Aerosol Mass Spectrometer (mAMS) to identify regions in cloud that may have been influenced by cloud seeding. Following sampling using a counter flow virtual impactor (CVI) that removes hydrometeors and other aerosol with aerodynamic diameters smaller than a certain size (~7 um in this study), cloud droplets are evaporated and the residual aerosol are tested for the presence of chemical species, including potassium (K) and Chlorine (Cl) that are otherwise not present (or not detectable in natural form) within the natural cloud environment. The authors (correctly) state that unambiguously identifying cloudy regions in which seeding material is contained within hydrometeors remains elusive; and developing a method to identify such regions is a key step to evaluating hypotheses associated with how cloud seeding might impact cloud and precipitation evolution. In this manuscript, the authors present evidence that they are able to do just that, and thus I believe this work makes an important contribution and is therefore worthy of publication. However, there are numerous points (some major) that I feel the authors need to address before publication.

**Response:** Authors thank the reviewer for the helpful and constructive comments on the manuscript. We have revised the manuscript thoroughly for better readability and simplicity.

In the first case study (case I, on Aug 21); the authors obtain measurements upwind in a stratus cloud and follow that with a release of seeding material a bit downwind, which is again followed with additional passes downwind to measure the cloud after seeding. The data shown in Figure 5 (b and c) indicate that most of the measurements downwind of the seeding release show elevated K and Cl. That is somewhat counter to what I would expect. The seeding material should disperse downwind in a slantwise pattern…similar to that shown in Tessendorf et al (2019; BAMS) and other references coming out of SNOWIE. Thus, when flying the 3 passes downwind in this case, I would expect the aircraft to fly through a brief line (region) containing seeding material surround by most of those legs being outside of regions containing seeding material. The authors could develop a simple advection model utilizing the mean wind speed and the time of the release along the seeding leg to estimate how the seeding line advects downwind. That would then provide an estimate of where to (and where not to) expect the presence of seeding material in the cloud.

**Response:** Thank you for the interesting discussion. Figure 5 (b,c) did not earlier display the spatial distribution of K and Cl; they were plotted with respect to Nt and re. Since aircraft measurements are confined to a limited area, the present data is insufficient to fully explain the spatial distribution of the plumes. Nevertheless, as suggested, we utilized a simple advection model to understand the transportation of the seeding agents in the downwind direction. It's worth noting that we used reanalysis wind data at a 0.25 x

0.25°(https://cds.climate.copernicus.eu/) resolution at an altitude of ~4 km on August 21, 2019, which was further resampled to 0.125 x 0.125°.

As illustrated in the figure below, the research aircraft conducted measurements within the seeding plumes in the downwind direction, after the dispersion of seeding agents.

Now the Figure 5 (shown as Fig. 1B here) is revised as suggested by the reviewer for clarity and better presentation of data.

[Figure]

Fig 1A:The red solid line represents the aircraft's track, while the black line represents the moment when the seeding agents were released. Advections of the seeding agents are shown in the downwind direction approximately 3 minutes after the seeding event. Wind data at a resolution of 0.25° x 0.25° is obtained from https://cds.climate.copernicus.eu/ and is further resample to create data at a resolution of 0.125° x 0.125°.

[Figure]

Fig. 1B: (a) Flight path during the seeding experiment on 21 August 2019. Liquid water content (LWC, gm$^{-3}$ ) at 1 Hz resolution is indicated. Periods during which cloud measurements were made for non-seeded cloud (NSCl) and seeded cloud (SCl) are indicated.  Mass concentrations (µg m$^{-3}$) obtained from the mAMS of (b) K and (c) Cl during the seeding experiment are shown along the flight track. The spatial background winds shown are obtained from https://cds.climate.copernicus.eu/  (0.25ºX0.25º), which are reampled to 0.125º X 0.125º.

That the authors are able to detect the presence of K and Cl is exciting, however, I think the onus is on them to prove that it is related specifically to the region that contains seeding material. My gut instinct is that it indeed is identifying those regions, but I am still a bit concerned based on the measurements provided in figure 5 as I describe in the above paragraph, that the method 'over-identifies' regions containing seeding material. I also wonder, since the same aircraft is used for releasing seeing material and making measurements; is there any potential for contamination? Were any experiments conducted where NSCl was resampled upwind following the release of seeding material?

**Response:**We could sample the seeding clouds downwind, after the dispersal of seeding agents in the cloud.

Since the stratus cloud usually covers a large area, assuming spatial uniformity in the cloud properties, the measurement of the seeded clouds appears un-contaminated due to the transect made by the aircraft.

Major Comments

1. Introduction—I think the introduction could be stronger. For example, in the second and third paragraphs the authors discuss the use of tracer materials to identify how researcher might understand the transport of seeding material through clouds. They describe two technologies previously used (chaff and chemical tracer such as SF6); while describing downsides of both, I think they miss the main point that neither of those methods would allow investigators to determine if seeding material actually makes it into cloud hydrometeors. This is touched on around line 85 ('can changes is cloud microphysical processing be linked to seeding materials'); in order for this to happen, the seeding material must actually interact with hydrometeors. The method developed in this manuscript does exactly that, by sampling residuals from cloud droplets. The author's should highlight this

**Response:** Thank you for the valuable suggestion. Now a few sentences are added emphasizing the limitation of past techniques.

2. Lines 131 – 139; and Table 1: Measurements of Cloud properties. The authors need some additional discussion on uncertainties in their measurements. These should come from published literature and simply referencing the manufacturer website is not sufficient. For example, the CDP provides a measurement/estimate of the size distribution, but there are uncertainties associated both with counting (though likely small) and sizing (which could be significant in some size bins)…these errors/uncertainties can lead to further (and much larger) uncertainties in integrated quantities such as liquid water content. The authors use LWC as a threshold in this study to identify regions less likely to be impacted by entrainment/mixing; thus having a discussion of potential errors in this measurement is important.

**Response**: The uncertainties related to measurements with single particle optical spectrometers, particularly from the CDP and CIP have been quantified and documented in multiple refereed publications. We now state these uncertainties in the modified text along with the appropriate references as written below.

"The uncertainties associated with the CDP, and single particle light scattering instruments like the CDP, have been well characterized and documented (Baumgardner et al., 1983, 2001, 2016; Lance et al., 2010). In water droplets the sizing uncertainty is ±20% and counting accuracy ±16%, which propagates into a LWC uncertainty of ±38%."

Limitation of selecting adiabatic fraction in the seeded cloud is now discussed in the revised manuscript. Please see section 2.1.

3. It appears that in the three cases presented, the authors knew well the cloud base conditions. Lines 144-5; the authors describe a threshold method to identify parcels that are either not or only slightly diluted. Why use LWC_max rather than LWC_ad? A threshold of 0.75 * LWC_max could be a highly diluted if the maximum LWC during that penetration is significantly less than the adiabatic value. Using adiabatic can be a direct estimate of 'how diluted' a parcel may be.

**Response:** We thank the reviewer for the valuable suggestion. Here, the main goal was to select the DSDs of higher LWC, which in general located in the core region of the cloud transect. The adiabatic LWC values are always greater than the maximum LWC measured in a cloud due to entrainment and mixing processes. In the present study, the adiabatic fraction i.e. $LWC_{max}/LWC_{ad}$ varies between 13% to 96%, which provides some idea of the degree of dilution. This means even in the cloud core entrainment of dry air take place lowering the Maximum LWC value. However, it may be noted that after seeding the distribution of LWC profile changes significantly compared to the non-seeded cloud. Parcel model simulations (without consideration of entrainment mixing process) of seeded and non-seeded cloud suggest changes in the distribution of LWC profiles (please see Figure 7 of Konwar et al., 2023). It is due to the reason that the supersaturation (SS) values at the lower altitude decreases in a seeded cloud (please see Figure 5 of Konwar et al., 2023). It is due to the reason that the coarse-mode aerosols nucleate and cloud droplets grow bigger consuming the available water vapor lowering the SS values. This means that the LWC profile in a seeded cloud would change due to the activation of coarse mode aerosols at the lower altitude. This would result changes in the adiabatic fraction of a seeded cloud. Therefore adiabatic fraction may not be the only parameter to understand the rate of dilution in a seeded cloud. This requires further detail model study with observational data. Limitations of using selecting the DSDs in the slightly diluted region are now discussed in section 2.1.

4. Specific for Seed Case i: Fig 5, Table 3, and paragraph contained in lines 329-343 – Several questions come up when I look at the presentation of this case: How are the incloud times selected for which dots are represented in figs 5b and 5c? I don't think I'd call these 'profiles' in fig5 b & c because all thee downwind (SCl) passes are at one level and the theNSCl pass is at a slightly higher level. I understand wanting a consistency in figure style between this case and the other two; but presenting this case in this style doesn't make sense to me. The data in table 3 indicates that one of the NSCl cases in this case had a LWC of 0.003 +/- 0.003 – this seems marginal as a cloud to me; less than 0.01 LWC is below what most would consider in cloud. It

seems odd to include data such as this as a comparison of cloud? LWCs are generally higher after seeding (although the measurements are lower in cloud…why? Seeding should not result in increased condensed water, but rather just changes in how that condensed water is distributed across hydrometeor distributions. On line 340 the authors suggest that increase in mid- sized droplet concentration may be due to collection, but this does not make sense…these droplets are <20 μm collection is extremely inefficient at these.

**Response**: Thank you for the valuable suggestions.

We agree the cloud pass had very small LWC values. We now have selected a new NSCl pass of nearly 30 sec with higher LWC than the previous case.  As suggested the word 'profile' is removed now for this cloud.

As suggested, the presentation for this seeding case is now changed for clarity.

5.  For Seed Case ii: In Fig 6 b and c – Need a better way to differentiate between which dots represent NSCl and which represent SCl – perhaps shape? One could be square— the other circle? Line 392 – the authors claim that after seeding Nt increased at lower altitudes—I disagree. The lowest altitude above cloud base sampled before seeding was 0.8 km and after seeding was 0.3 km – these two cannot be compared---the so seed is more than twice the distance above cloud base. Further, in the after seeding case, a pass was made at 0.96 km, and that contained lower concentrations than the no seed at 0.8 km. The authors need more rigor in their data analysis before making such claims. On line 398 the authors claim that large standard deviations in re are likely the result of entrainment/mixing with dry air….I agree; but then they need to be more rigorous in trying to eliminate regions which are influenced by mixing in their analysis. This is a possible indicator that their threshold of 0.75*LWC_max is not sufficient to do accomplish this. On line 412 the authors suggest that because drizzle is encountered after seeding and not before that the production of drizzle can be attributed to seeding…while this is one possibility, there are many other possibilities that could account for that. Different stages of lifecycle, pre-conditioning of the environement, etc are all possibilities. Studies of cumulus continually show that drizzle may form at the same levels in a cumulus cluster that indicated no drizzle development several minutes earlier.

**Response:**As suggested the symbols are improved and now distinguishable for SCl and NSCl cases.

We have revised the discussion keeping in mind the limited data and a statement is included on the small data volume. We already had acknowledged the possible role of natural variability on drizzle production.

6. For Seed Case iii: Why is the LWC 3-5 times higher after seeding than before, when observations were made at the same level above cloud base? One possibility is that these difference are due solely to difference in amount of dry air mixed into parcels that were sampled, and any differences in microphysics may be attributed to that rather than to seeding?

**Response:** Thank you for pointing out the important aspect. Now we have done additional analysis to indicate the role of strong updrafts in these clouds. A discussion is now added to the revised manuscript: "Another aspect to consider here is the effect of strong updraft of 8 m s$^{-1}$. Using the Twomey (1959) equation the maximum droplet concentration formed in an updraft (W) can be expressed in terms of W and CCN-SS spectra, i.e. $N_{CCN}$=C SS$^k$ i.e. (Roger and Yau, 1989),

$$N \approx 0.88\ C^{2/(k+2)}\left[7\ X\ 10^{-2}\ W^{3/2}\right]^{k/(k+2)} \tag{4}$$

Here, W is in cm s$^{-1}$, $N_{CCN}$= 799 SS$^{0.43}$, which is obtained from the CCN counter (Roberts and Nenes, 2005; Nenes et al., 2001 and reference therein) operated in the research aircraft. During the cloud passes, maximum updrafts of W= 2.89 m s$^{-1}$, 1.00 ms$^{-1}$ and 1.91 m s$^{-1}$ were obtained. These values suggest that droplets formed in these updrafts could be 593 cm$^{-3}$, 448 cm$^{-3}$ and 531 cm$^{-3}$, respectively. If we use the maximum updraft speed of 8 m s$^{-1}$ measured below cloud base, the droplet concentrations formed in this updraft could be as high as 777 cm$^{-3}$. Therefore, the presence of strong updrafts that yield high SS could be one reason for the increasing $N_t$ in the seeded clouds; while dry air mixing in the NSCl cases could be another reason for the smaller concentration of $N_t$. "

Specific Comments

1. Abstract, Line 31 – ….attempts to link precipitation enhancement….remained inclusive? Not sure what is meant by this, could it be the authors mean that attempts remained elusive?

**Response:** Yes now the word 'elusive' is used.

2. Abstract, Line 37 -- ….cloud droplets underwent a drying process, … I think it would be more accurate to state that droplets were evaportated.

**Response:** We agree, now it is revised.

3. Line 67 – '…then these traces are tried to measure higher in the cloud.' Wording here is incorrect.

**Response:** OK, now the sentence is revised.

4. Line 91 – '…to identify seeding material in the residual cloud droplets.' The seeding material is identified within 'cloud droplet residuals'; i.e. the aerosol that remains after evaporation of the cloud droplets.

**Response:** OK, now the sentenced is revised.

5. Line 92 – '….hypothesis relies on a chain of microphysical mechanisms.' The wording here is not correct; there is no such thing as a chain of mechanisms. I think the authors are referring to a chain of events with each event being a specific microphysical process.

**Response:** We referred to the chain of microphysical processes. Now the sentence is revised.

6. Line 99 – 'CCN do..' (not does, CCN is plural)

**Response:** OK, now the sentence is corrected.

7. Line 107 – replace 'seed' with 'seeding agent' and delete the word 'the' before broadening.

**Response:** OK, now corrected.

8. Sentence on lines 111-113 – I read this sentence in relation to the evolution of microphysical processes resulting from hygroscopic seeding (previous sentence); however, this has been accomplished in glaciogenic seeding experiments as the authors reference French et al. (2018). It seems to me that the processes are more dynamic and complex in cold clouds, yet that linkage has been made.

**Response:** This reference was included to emphasize understanding of physical processes of cloud seeding experiment, in general. The reviewer is correct, now this sentence is precluded from the revised manuscript.

9. Lines 127-129 – isn't all the data presented in this study from 'warm' cloud seeding? Why reference flare racks for 'cold' cloud seeding. And, by 'warm' and 'cold' are the authors referring to hygroscopic vs glaciogenic seeding? If so, they should be referred to as such.

**Response:** Two relevant references are cited in the revised manuscript.

10. Use of abbreviations and nomenclature for height of measurements in cloud— introduced on Lines 146 & 7 – The authors use the nomenclature 'cloud depth' (abbreviated CD) to describe the heights (above cloud base) for measurements throughout the manuscript. I find this confusing as 'cloud depth' normally refers to a measurement of the total depth (or thickness) of a cloud. I suggest the authors should change all instances in text, figures, and tables from 'cloud depth' to 'height above cloud base'.

**Response:** Ok, as suggested we now used the term D* (km) for vertical distance above the cloud base.

11. Line 161 – Delete 'As mentioned earlier'

**Response: OK.**

12. Line 165 & 6 "…the CVI inlet segregates and samples cloud elements." Replace that with: 'the CVI inlet cloud hydrometeors with aerodynamic diameters larger that a certain size depending the rate of the counter flow.

**Response: OK.**

12. Line 165 & 6 "…the CVI inlet segregates and samples cloud elements." Replace that with: 'the CVI inlet cloud hydrometeors with aerodynamic diameters larger that a certain size depending the rate of the counter flow.

**Response:** OK.

13. Line 168 – 'larger than >7 µm' ' how was this verified for the setup in this experiment? The cutoff size is dependent not only on flow rate, but also aircraft speed, possibly mounting location on the aircraft, altitude, etc. Verifying this (somehow) seems to be a critical thing in this experiment.

**Response:** The flow rates of the CVI were verified using a Gilibrator flow meter as the standard before the project started. The CVI adjusted flow rates with its internal software based on true air speed (TAS) provided to the CVI computer from the AIMMS probe. The AIMMS measured true air speed and the M300 *acquisition system* sent this out to the CVI once per second and flow rates were adjusted accordingly. Flow rates were monitored and verified that they were being kept at the set points. Brechtel Engineering, the provider of the CVI, has characterized the cut-size as a function of air pressure and air speed. The angle of the CVI was determined from the average angle of attack. The reviewer is correct that a number of factors determine this cut size and the text has been modified to state that this cut-size is known to approximately +/- 1 um. This has little impact on the results and conclusions.

In addition to the above point we want to state that we have now corrected the mAMS data for enhancement factor as the ambient aerosol concentration concentrated in the CVI tip (Shingler et al., 2012). A sentence on this aspect is now added page 15.

14. Line 172 – I cannot access the Golderger et al. (2020) reference.

**Response:** Thank you for pointing this out. The following link is now provided in the reference list: https://www.arm.gov/publications/tech_reports/handbooks/doe-sc-arm-tr-254.pdf

15. Lines 230 & 231 and Figure 3—Is 'start' and 'stop' the times of 'Pass Begin' and 'Pass End' shown in the figure? If so, be consistent in labeling. Label 'maximum total mass time' in figure 3.

**Response:** OK, now corrected.

16. Lines 253 through 260 – The authors repeat items that are already discussed earlier and restate the same things multiple times in lines in this paragraph. I think the authors should rewrite this paragraph to make it more concise and less repetitive.

**Response:** This paragraph is now revised for clarity.

17. Line 285 and 286 reword to: …'with three cloud passes of the same cloud before and three passes after seeding are shown in Fig 4.'

**Response:** OK.

18. Line 288 – change 'above' to 'compared to'

**Response:** OK.

19. Line 290 – the wording seem awkward here…maybe change 'consist of' to 'containing'

**Response:** OK.

20. Line 291 & 2 – I do not see increased SO4 concentrations in the seeded clouds (perhaps at mid level, but certainly not at the highest level…

**Response:** Now this sentence is revised, mentioning mid level.

21. Line 307 – 'several cloud passes….were made….before dispersal of seeding material.' From Figure 5a, it looks like one pass was made.

**Response:** Now the sentence is revised.

22. Line 311-314 – The AIMMS 20 does not measure whether in cloud, I assume the authors use some threshold on the CDP measurements? Possibly a threshold on the LWC derived from the integrated size distribution?

**Response:** Now it is mentioned, $N_t > 10$ cm$^{-3}$ and passes greater or equal than 5 sec was the criteria.

23. Line 326 – sentence that begins 'Discussions on cloud probes…' ---delete this.

**Response**: OK.

---

## Author Response (AR1)

Date: 22-11-2023

Dear Dr. Editor,

Atmospheric Measurement Techniques,

We are happy to submit the revised manuscript entitled '**Identifying the seeding signature in cloud particles from hydrometeor residuals'** to the prestigious journal *Atmospheric Measurement Techniques* for a possible publication.

We now have incorporated all the suggestions made by the two anonymous reviewers. With this revision we hope that the manuscript will be accepted for publication in AMT

Sincerely yours,

Thanking you,

Dr. Mahen Konwar

Indian Institute of Tropical Meteorology, Pune, India

**Responses to Reviewer#1**

This well-written study shows that the seeding agent in hygroscopic flares can be detected in the cloud droplets, and related to the indicated changes in cloud microstructure.

The indicated effect of the flares in the limited sample is mainly the tail effect of the largest particles initiating large cloud drops and drizzle. These results do not support the competition effect, although much more data is required for conclusive evidence. I encourage the authors to add such a statement in their conclusions.

**Response**: We thank the reviewer for the constructive and helpful suggestions.

Now a statement is included to emphasis the importance of the tail effect, indicating the more data are required for conclusive evidence. In addition, we also observed the role of strong updrafts in the activation of small sized CCNs in the isolated convective clouds. Now these discussions are included in the revised manuscript. The complexity in the cloud seeding experiment is stated.

There are several minor comments:

Line 31: Change "inclusive" to "elusive".

**Response**: OK, it is changed now.

Figure 6b and c; Fig 7b and c: It is impossible to separate the SCl and NSCl cloud segments. If the yellow points are considered NSCl, please state so explicitly and justify it.

**Response**: As suggested the images are now changed for clarity.

---

## Referee Report (RR1)

Review of 'Identifying the seeding signature in cloud particles from hydrometeor residuals'

Konwar et al.

AMT-2023-171

This is a second review following my first review of the initially submitted manuscript. As stated in my initial review, I believe this work makes an important contribution and is worthy of publication. However, I also noted several major points that the authors needed to address prior to publication. Unfortunately, I do not feel my points have been addressed adequately in the revised manuscript and/or in the 'response to reviewer' comments. For this reason, I recommend rejecting the manuscript at this time.

Below I point to specific comments in my original review and describe how/why I believe the authors response was not adequate in many cases.

In my summary review, second paragraph, I noted that I did not understand why all of the regions downwind appeared to have elevated K and Cl. The authors responded by noting that my original interpretation was incorrect and revised Figure 5 to illustrate elevated levels were only sampled at discrete locations. This was an excellent answer an explanation to my original question.

In the last two sentences of the third paragraph of my summary review, I asked the following questions: "I also wonder, since the same aircraft is used for releasing seeing material and making measurements; is there any potential for contamination? Were any experiments conducted where NSCl was re-sampled upwind following the release of seeding material?"
The authors responded: "We could sample the seeding clouds downwind, after the dispersal of seeding agents in the cloud.
Since the stratus cloud usually covers a large area, assuming spatial uniformity in the cloud properties, the measurement of the seeded clouds appears un-contaminated due to the transect made by the aircraft."
This does not answer the question(s) asked. Also, the authors state they can 'assume spatial uniformity' in cloud properties, but the LWC color plot in Fig 5a clearly indicates a lot of heterogeneity, even in regions with no elevated K or Cl. Also, in the new Fig 5, there is clearly a small area of elevated K and Cl prior to seeding release. The authors should explain that.

 Under Major Comments

**1 – I suggested that the authors could strengthen the introduction by noting that previous tracer technologies used to identify seeded regions were unable to determine if the seeded material actually makes it into hydrometeors. (This new methodology does just that!).**
The authors responded: "Thank you for the valuable suggestion. Now a few sentences are added emphasizing the limitation of past techniques."

However, I could not find where this had been added. If the author's had provided a revised manuscript with changes tracked (or even provided the new text in their response) it would have been helpful.

**3 I asked the question, that if the authors wanted to ensure they were in the 'core' of the cloud why use LWC_max instead of LWC_ad?**
The author response (not copied here for brevity) stated that the goal was to select 'core' regions of the transect, noting (correctly) that measured LWC is always less than adiabatic values due to entrainment and mixing. They went on to state that in the present study ratios of max to adiabatic varied between 13 to 96%. I would suggest, that if they included regions of cloud in this study in which the maximum LWC was only 13% of adiabatic, than changes in cloud microstructure would be *strongly influenced* by processes such as entrainment and mixing. It is incumbent on the authors to prove that changes/differences in microstructure between seeded and non-seeded clouds are the result of seeding and not some other (natural) process.

**4 I asked the question(s) at the end of the comment: LWCs are generally higher after seeding (although the measurements are lower in cloud)...why? Seeding should not result in increased condensed water, but rather just changes in how the condensed water is distributed....etc.**
The authors response did not address this question.

**6 I asked roughly the same question as in #4, except for seeding case iii.**
The authors responded describing why Nt might be different and discussed the role of updraft; all of which I agree with, but disregards the question completely.

---

## Author Response (AR2)

**Response to Reviewer#2**

Review of 'Identifying the seeding signature in cloud particles from hydrometeor residuals' Konwar et al. AMT-2023-171

This is a second review following my first review of the initially submitted manuscript. As stated in my initial review, I believe this work makes an important contribution and is worthy of publication. However, I also noted several major points that the authors needed to address prior to publication. Unfortunately, I do not feel my points have been addressed adequately in the revised manuscript and/or in the 'response to reviewer' comments. For this reason, I recommend rejecting the manuscript at this time.

Below I point to specific comments in my original review and describe how/why I believe the authors response was not adequate in many cases.

**Response:** The authors never intended to ignore the suggestions provided by the esteemed reviewer. The constructive comments helped immensely to improve the quality of the manuscript. Below are the clarifications given as a response to the points raised by the reviewer.

In my summary review, second paragraph, I noted that I did not understand why all of the regions downwind appeared to have elevated K and Cl. The authors responded by noting that my original interpretation was incorrect and revised Figure 5 to illustrate elevated levels were only sampled at discrete locations. This was an excellent answer an explanation to my original question

**Response:** We think there is a misunderstanding; we never questioned the reviewer's suggestion. We in fact (as suggested by the reviewer) used a simple advection model to understand the locations of the seeding particles in the downwind direction. Accordingly, the texts and figures were revised. However, since this study did not consist of a component with numerical simulation, we did not include the advection model results in the manuscript but did so in the response to the reviewer. In the revised manuscript, we have included a discussion on the advection of the plumes in the slanted downwind direction. Advection of seeding plumes as reported from CAIPEEX experiment (Gayatri et al., 2023) and the SNOWIE field program (Xue et al., 2022) is now discussed in the revised manuscript. Gayatri et al., 2023 illustrated the seeding impact downwind of the seeded area through the high-resolution numerical model in the same monsoon environment with the monsoon low-level jet (LLJ) as detailed in the present study. The cloud bases are situated very close to the region with high wind speeds in the monsoon low-level jet and the advection of seeding plume downwind of the seeded location is noted. However, the fact that seeding was done specifically in the strong updraft zones and the seed particles were also lifted inside the cloud and more cloud droplets were noted both in the observations and simulations. The clouds selected for seeding had higher liquid water content (Prabhakaran et al., 2023).

**Reference:**

Gayatri K, Prabhakaran T., Malap N., Konwar M., Gurnule D., Bankar S., Murugavel P. Physical evaluation of hygroscopic cloud seeding in convective clouds using in situ observations and numerical simulations during CAIPEEX, (2023), *Atmospheric Research*, 284: 106558, March 2023, DOI:10.1016/j.atmosres.2022.106558, 1-17

Prabhakaran T., Murugavel P., Konwar M., Malap N., Gayatri K., Dixit S., Samanta S., Chowdhuri S., Bera S., Varghese M., Jaya Rao Y., Sandeep J., Safai P.D., Sahai A.K., Axisa D., Karipot A., Baumgardner D., Werden B., Fortner E., Hibert K., Nair S., Bankar S., Gurnule D., Todekar K., Jose J., Jayachandran V., Soyam P.S., Gupta A., Choudhary H., Aravindhavel A., Kantipudi S.B., Pradeepkumar P., Krishnan R., Nandakumar K., DeCarlo P.F., Worsnop D., Bhat G.S., Rajeevan M., Nanjundiah R., CAIPEEX - Indian cloud seeding scientific experiment, *Bulletin of the American Meteorological Society*, 104, November 2023, DOI:10.1175/BAMS-D-21-0291.1, E2095–E2120

Xue L., C. Weeks, S. Chen, S. A. Tessendorf, R. M. Rasmussen, K. Ikeda, B. Kosovic, D. Behringer, J. R. French, K. Friedrich, T. J. Zaremba, R. M. Rauber, C. P. Lackner, B. Geerts, D. Blestrud, M. Kunkel, N. Dawson, and S. Parkinson (2022), Comparison between Observed and Simulated AgI Seeding Impacts in a Well-Observed Case from the SNOWIE Field Program
Journal of Applied Meteorology and Climatology, page345–367, https://doi.org/10.1175/JAMC-D-21-0103.1

In the last two sentences of the third paragraph of my summary review, I asked the following questions: "I also wonder, since the same aircraft is used for releasing seeing material and making measurements; is there any potential for contamination? Were any experiments conducted where NSCl was re-sampled upwind following the release of seeding material?" The authors responded:

"We could sample the seeding clouds downwind, after the dispersal of seeding agents in the cloud. Since the stratus cloud usually covers a large area, assuming spatial uniformity in the cloud properties, the measurement of the seeded clouds appears un-contaminated due to the transect made by the aircraft."

This does not answer the question(s) asked. Also, the authors state they can 'assume spatial uniformity' in cloud properties, but the LWC color plot in Fig 5a clearly indicates a lot of heterogeneity, even in regions with no elevated K or Cl. Also, in the new Fig 5, there is clearly a small area of elevated K and Cl prior to seeding release. The authors should explain that.

**Response**: We apologize for this oversight. The aircraft could indeed release non-volatile and fine aerosol particles through exhaust emissions (Anderson et al., 1998), which may also contaminate the cloud mass. Further, Prabhakaran et al. (2023) also compared flare size distribution with the background and aircraft exhaust aerosol size distributions (see, supplementary material https://doi.org/10.1175/BAMS-D-21-0291.2). That study indicates the potential impact of aircraft exhaust on the ambient aerosol size distributions that might alter mean radius and number

concentrations with different modes of log-normal size distributions. A similar discussion is now added in the revised manuscript, please see page 18.

The reviewer is correct in pointing out the heterogeneity (did not mention in the manuscript) of the cloud structure. This oversight on the cloud structure has been corrected.

The reviewer is also correct in pointing out a small area of elevated K and Cl, prior to the flare burning. This was measured outside the cloudy. It might be appeared probably due to other unknown sources. Now the same is mentioned in the figure caption.

Under Major Comments

**1 – I suggested that the authors could strengthen the introduction by noting that previous tracer technologies used to identify seeded regions were unable to determine if the seeded material actually makes it into hydrometeors. (This new methodology does just that!). The authors responded:**

"Thank you for the valuable suggestion. Now a few sentences are added emphasizing the limitation of past techniques."

However, I could not find where this had been added. If the author's had provided a revised manuscript with changes tracked (or even provided the new text in their response) it would have been helpful.

**Response**: We apologize for the confusion. The sentence 'Using these tracers as proxies for tracking air masses carrying seeding material is limited by the challenge of unambiguously connecting their presence with the seeding material due to their non-reactive nature with cloud particles.' Now it is added in the introduction section.  For reference, now it is highlighted with red color in the track changed version of the manuscript. Please see L72-74, page 3-4.

**3 I asked the question, that if the authors wanted to ensure they were in the 'core' of the cloud why use LWC_max instead of LWC_ad? The author response (not copied here for brevity) stated that the goal was to select 'core' regions of the transect, noting (correctly) that measured LWC is always less than adiabatic values due to entrainment and mixing. They went on to state that in the present study ratios of max to adiabatic varied between 13 to 96%. I would suggest, that if they included regions of cloud in this study in which the maximum LWC was only 13% of adiabatic, than changes in cloud microstructure would be strongly influenced by processes such as entrainment and mixing. It is incumbent on the authors to prove that changes/differences in microstructure between seeded and non-seeded clouds are the result of seeding and not some other (natural) process.**

**Response:** The revised figures include ranges of LWC adiabatic fractions, along with the LWC/LWC$_{max}$ ranges. In addition, we have included descriptions highlighting the impact of the entrainment and mixing process on our results. We also noted that the adiabatic fraction values could be highly variable near the clouds base. For example near the cloud base, the LWC values are quite small, e.g. <1 g m$^{-3}$, a small change in the measured LWC values could yield a large change in adiabatic fraction (AF). Other aspects, for example, the effect of seeding particles on the LWC profiles may also affect the AF, the

effect of drizzle formation in the cloud can decrease AF are discussed in the manuscript. Please refer to page 7-8, L 160-163 of the revised manuscript (please see the track changed version).

An example of the variation of LWC non-seeded cloud (NSCl), seeded clouds (SCl) and adiabatic LWC (LWC$_{ad}$) with respect to the distance from the cloud base (D*, km) is shown in the figure below.The case considered is for 23$^{rd}$ August 2019. The LWC for NSCl and SCl cases are calculated in the size range of 3-50 µm.It can be seen that the LWC values of both NSCl and SCl are smaller than the LWC$_{ad}$ values at all D*. Two SCl cloud passes near the cloud base were less diluted; however, uncertainty remains as the release of seeding materials may activate new cloud droplets and change the LWC values. At higher D*, LWC values are quite small as cloud droplets convert into drizzle drops.

[Figure]

Figure: Case 23$^{rd}$ August 2019. Profile of non-seeded cloud (NSCl) LWC and seeded cloud (SCl) LWC are shown with respect to the distance from the cloud base (D*, km). Standard deviations from the mean values are also shown.

**4 I asked the question(s) at the end of the comment: LWCs are generally higher after seeding (although the measurements are lower in cloud)…why? Seeding should not result in increased condensed water, but rather just changes in how the condensed water is distributed….etc. The authors response did not address this question.**

**Response:** We respectfully disagree with the reviewer that seeding cannot increase the LWC. Seeding will not change the adiabatic LWC, i.e. the potentially available LWC, however, by activating new droplets, this does increase the LWC because these new droplets have now converted the available water vapor into liquid water. Since these clouds under consideration are convective in nature and there are significant changes in LWC also due to entrainment and evaporation effects. Several of our previous studies (Prabha et al., 2011; Patade et al., 2015; 2019) have illustrated that more aerosols in the sub-cloud layer can increase LWC with height.

Reference:

Patade, S., Kulkarni, G., Patade, S., Deshmukh, A., Dangat, P., Axisa, D., Prabha, T. V.: Role of liquid phase in the development of ice phase in monsoon clouds: Aircraft observations and numerical simulations. Atmos. Res., 229, 157–174, 2019. https://doi.org/10.1016/j.atmosres.2019.06.022

Patade, S., Prabha, T. V., Axisa, D., Gayatri, K., Heymsfield, A,: Particle size distribution properties in mixed-phase monsoon clouds from in situ measurements during CAIPEEX Jour. of Geophys. Res. Atmos., 120, 19, 2015.

**6 I asked roughly the same question as in #4, except for seeding case iii. The authors responded describing why Nt might be different and discussed the role of updraft; all of which I agree with, but disregards the question completely.**

**Response:** Please see our response to comment #4. These are illustrated in the revised manuscript.

---

## Author Response (AR3)

This study demonstrates successfully that the dispersed elements of the seeding flares can be detected in concentrations well above the background in seeded clouds, and be compared to comparable (with respect to D* and AF) not seeded clouds. It also shows changes in the cloud and drizzle drop size distributions (DSD) which agree with the expected effects of hygroscopic flares. Therefore, this study should be published despite remaining major uncertainties in this study.

The major uncertainty is the dynamic evolution of the clouds between seeding and sampling. However, constraining the comparisons to comparable D* and AF is sufficient for relating the consistent differences in the DSD to the seeding effects. Since the flares produce large concentrations of very small aerosols and low concentrations of giant CCN, the expected effect of secondary activation along with added drizzle appears to be realized.

**Reply:** We thank the reviewer for the encouraging comments on the manuscript.

One minor correction is to the claim that secondary drop activation can increase AF. This is true only if the secondary drop activation decreases the in-cloud supersaturation by the amount of the added AF. It is unlikely that in clouds with drop concentration exceeding 80 cm$^{-3}$ and modest updraft the supersaturation would be greater than about 1% or 2% at most, which is much less than the observed fractional difference in AF between the seeded and not seeded clouds. Controlling carefully for the AF between segments of seeded and not seeded clouds can address the concern of a bias between the seeded and control clouds.

**Reply:** We thank the reviewer for the discussion on adiabatic fraction. The adiabatic fraction (AF) was calculated from the formula:  $AF=LWC/LWC_{ad}$, where LWC is the measured  liquid water content in a horizontal cloud pass. $LWC_{ad}$ is the theoretically calculated adiabatic liquid water content using a parcel model. $LWC_{ad}$  is constant at a given altitude. We stated in the manuscript that activation of new particles can change the AF values.  This can be proved from the following calculations:

Let us assume that initially total droplet concentration of the non-seeded cloud was $N_{tNSCl}$=10 cm$^{-3}$, and the corresponding $LWC_{NSCl}$= 1 g m$^{-3}$.  Since the seeding flares consist of small particles (Prabhakaran et al., 2023; Konwar et al. 2023), they may activate at a higher supersaturation in the cloud. The monsoon clouds typically have strong updrafts (exceeding 10 ms$^{-1}$) resulting in high quasi-state supersaturation (up to 5%-8%) that may activate small size aerosol particles in the convective clouds (Prabha, et al., 2011). When collision-coalescence process is active droplet concentration decreases which result in the increase in supersaturation values, and small particles can be activated. The same phenomenon was noted in a parcel model for seeded clouds when flare size distribution was used for simulation (e.g., Konwar et al., 2023).  After seeding let us assume that secondary nucleation took place which can increase total droplet concentrations, assuming $N_{tSCl}$=20 cm$^{-3}$ which will also increase LWC (let us assume that the increment is by an amount 0.50 g m$^{-3}$) in the seeded cloud. Therefore LWC of the seeded cloud is:  $LWC_{SCl}$= $LWC_{NSCl}$+ 0.5 g m$^{-3}$=1.5 g m$^{-3}$.

Since $LWC_{ad}$ remains constant at a given altitude, let us assume it as 2 g m$^{-3}$ for the present case.

Therefore the adiabatic fractions for no-seeded and seeded clouds are: $AF_{NSCI}$= 1/2=0.5 and $AF_{SCI}$= 1.5/2=0.75, which indicates that $AF_{SCI}$> $AF_{NSCI}$ . The activation of new particles can impact the in-cloud supersaturation, but is not likely to associate with the AF under discussion.

For more clarity on the aspect of nucleation of the small size particles, a few sentences have been added to the revised manuscript, : 'Since the cloud seeding flare produces high concentrations of small-sized particles, they can be activated into cloud droplets in strong updraft regimes with high supersaturation (Konwar et al., 2023; Prabhakaran et al., 2023). In a parcel model simulation, small aerosols released from flares are found to be activated due to an increase in supersaturation when the collision-coalescence process is active (Konwar et al., 2023). For details on the nucleation process within the zone of intense collision, where rapid decrease in drop concentration leads to an increase in supersaturation, readers are referred to Pinsky and Khain (2002).)'. Please see Page 7-8, L 160-167 of the revised manuscript.

**References:**

Konwar, M., Malap, N.,  Hazra, A., Axisa, D., Prabhakaran, T., and Khain, A.: Measurement of Flare Size Distribution and Simulation of Seeding Effect with a Spectral Bin Parcel Model, Pure and Applied Geophysics, 180, 3019–3034, 2023, https://doi.org/10.1007/s00024- 023-03293-z.

Pinsky, M., Khain, A. P.:  Effects of in-cloud nucleation and turbulence on droplet spectrum formation in cumulus clouds. Quart. J. Roy. Met. Soc., 128, 1-33, 2002.

Prabha, T. V., Khain, A., Maheshkumar, R. S., Pandithurai, G., Kulkarni, J. R., Konwar, M, and Goswami, B. N.: Microphysics of premonsoon and monsoon clouds as seen from *in situ* measurements during the Cloud Aerosol Interaction and Precipitation Enhancement Experiment (CAIPEEX), J. Atmos. Sci.,68, 1882–1901, 2011.

Prabhakaran, T., Murugavel, P., Konwar M., Malap, N., Gayatri, K., Dixit, S., Samanta, S., Chowdhuri., S., Bera, S., Varghese, M., Rao, J., Sandeep, J.,  Safai, P. D., Sahai, A. K., Axisa, D., Karipot, A., Baumgardner, D., Werden, B., Fortner, Ed, Hibert, K., Nair, S., Bankar, S., Gurnule, D., Todekar, K., Jose, J., Jayachandran, V., Soyam, P. S., Gupta, A., Choudhary, H., Aravindhavel, A.,  Kantipudi, S. B., Pradeepkumar, P., Krishnan, R., Nandakumar, K., DeCarlo, P. F.,  Worsnop, D., Bhat, G. S., Rajeevan, M., and Nanjundiah, R.: CAIPEEX - Indian cloud seeding scientific experiment ,  Bulletin of American Meteorological Society, 2023,  https://doi.org/10.1175/BAMS-D-21-0291.1